# FairPFN: A Tabular Foundation Model for Causal Fairness

**Jake Robertson** [1 2]   **Noah Hollmann** [3 4]   **Samuel Müller** [5 2]   **Noor Awad** [2]   **Frank Hutter** [4 1 2]

## Abstract

Machine learning (ML) systems are utilized in critical sectors, such as healthcare, law enforcement, and finance. However, these systems are often trained on historical data that contains demographic biases, leading to ML decisions that perpetuate or exacerbate existing social inequalities. Causal fairness provides a transparent, human-in-the-loop framework to mitigate algorithmic discrimination, aligning closely with legal doctrines of direct and indirect discrimination. However, current causal fairness frameworks hold a key limitation in that they assume prior knowledge of the correct causal model, restricting their applicability in complex fairness scenarios where causal models are unknown or difficult to identify. To bridge this gap, we propose FairPFN, a tabular foundation model pre-trained on synthetic causal fairness data to identify and mitigate the causal effects of protected attributes in its predictions. FairPFN's key contribution is that it requires no knowledge of the causal model and still demonstrates strong performance in identifying and removing protected causal effects across a diverse set of hand-crafted and real-world scenarios relative to robust baseline methods. FairPFN paves the way for promising future research, making causal fairness more accessible to a wider variety of complex fairness problems.

## 1. Introduction

Algorithmic discrimination is among the most pressing AI-related risks of our time, manifesting when machine learning (ML) systems produce outcomes that disproportionately disadvantage historically marginalized groups (Angwin et al., 2016). Despite significant advancements by the fairness-aware ML community, critiques highlight the contextual limitations and lack of transferability of current statistical fairness measures to practical legislative frameworks (Weerts et al., 2023). In response, the field of *causal fairness* has emerged, providing a transparent and human-in-the-loop causal framework for assessing and mitigating algorithmic bias with a strong analogy to existing anti-discrimination legal doctrines (Plecko & Bareinboim, 2024).

A recent review comparing outcome-based and causal fairness approaches (Castelnovo et al., 2022) argues that the non-identifiability of causal models from observational data (Pearl, 2009) limits the usage of current causal fairness frameworks in practical applications. In practice, users must provide full or partial information about the underlying causal model, a challenging task given the complexity of systemic inequalities. Furthermore, an incorrectly presumed causal graph, such as one falsely assuming a variable is independent of a protected attribute, can invalidate causal fairness metrics (Ma et al., 2023; Binkytė-Sadauskienė et al., 2022), resulting in *fairwashing* and fostering a false sense of security and trust.

This paper takes a bold new perspective on achieving causal fairness. Our **key contribution** is FairPFN, a tabular foundation model for causal fairness, pre-trained on synthetic causal fairness data to learn to identify and remove the causal effects of protected attributes in tabular classification settings. When used on a new dataset, FairPFN does *not* rely on a user-specified causal model or graph, instead solely relying on the causally-generated data it has seen during pre-training. We demonstrate through extensive experiments that FairPFN effectively and consistently mitigates the causal impact of protected attributes across various hand-crafted and real-world scenarios, yielding causally fair predictions without user-specified causal information. We summarize our various contributions:

1. **PFNs for Causal Fairness** We propose a paradigm shift for algorithmic fairness, in which a transformer is pre-trained on synthetic causal fairness data.

2. **Causal Fairness Prior:** We introduce a synthetic causal data prior which offers a comprehensive representation for fairness datasets, modeling protected attributes as binary exogenous causes.

---

[*]Equal contribution  [1]ELLIS Institute Tübingen [2]University of Freiburg [3]Charité University Medicine Berlin [4]Prior Labs [5]Meta. Correspondence to: Jake Robertson <robertsj@cs.uni-frieburg.de>.

*Proceedings of the 42nd International Conference on Machine Learning*, Vancouver, Canada. PMLR 267, 2025. Copyright 2025 by the author(s).

3. **Foundation Model:** We present FairPFN, a foundation model for causal fairness which, given only observational data, identifies and removes the causal effect of binary, exogenous protected attributes in predictions, and demonstrates strong performance in terms of both causal fairness and predictive accuracy on a combination of hand-crafted and real-world causal scenarios. We provide a prediction interface to evaluate and assess our pre-trained model, as well as code to generate and visualize our pre-training data at `https://github.com/jr2021/FairPFN`.

## 2. Related Work

In recent years, causality has gained prominence in the field of algorithmic fairness, providing fairness researchers with a structural framework to reason about algorithmic discrimination. Unlike traditional fairness research (Kamishima et al., 2012; Agarwal et al., 2018; Hardt et al., 2016), which focuses primarily on optimizing statistical fairness measures, causal fairness frameworks concentrate on the structure of bias. This approach involves modeling causal relationships among protected attributes, observed variables, and outcomes, assessing the causal effects of protected attributes, and mitigating biases using causal methods, such as optimal transport (Plecko & Bareinboim, 2024) or latent variable estimation (Kusner et al., 2017; Ma et al., 2023; Bhaila et al., 2024).

Counterfactual fairness, introduced by Kusner et al. (2017), posits that predictive outcomes should remain invariant between the actual world and a counterfactual scenario in which a protected attribute assumes an alternative value. This notion has spurred interest within the fairness research community, resulting in developments like path-specific extensions (Chiappa, 2019) and the application of Variational Autoencoders (VAEs) to create counterfactually fair latent representations (Ma et al., 2023).

The initial counterfactual fairness framework necessitates comprehensive knowledge of the causal model. In contrast, the Causal Fairness Analysis (CFA) framework (Plecko & Bareinboim, 2024) relaxes this requirement by organizing variables within a Standard Fairness Model (SFM) for bias assessment and mitigation. Moreover, the CFA framework presents the Fairness Cookbook, which defines causal fairness metrics—Indirect-Effect, Direct-Effect, and Spurious-Effect—that directly align with US legal doctrines of disparate impact and treatment. Furthermore, the CFA framework challenges Kusner et al. (2017)'s modeling of protected attributes as exogenous causes, permitting correlations between protected attributes and confounding variables that contribute to the legally admissible Spurious-Effect.

## 3. Background

This section establishes the scientific foundation of FairPFN, including terminology relevant to algorithmic fairness, causal ML, counterfactual fairness, and prior-data fitted networks (PFNs).

**Algorithmic Fairness** Algorithmic discrimination occurs when historical biases against demographic groups (e.g., ethnicity, sex) are reflected in the training data of ML algorithms, leading to the perpetuation and amplification of these biases in predictions (Barocas et al., 2023). Fairness research focuses on measuring algorithmic bias and developing *fairness-aware* ML models that produce non-discriminatory predictions. Practitioners have established over 20 fairness metrics, which generally break down into group-level and individual-level metrics (Castelnovo et al., 2022). These metrics can be used to optimize predictive models, balancing the commonly observed trade-off between fairness and predictive accuracy (Weerts et al., 2024).

**Causal Machine Learning** Causal ML is a developing field that leverages modern ML methods for causal reasoning (Pearl, 2009), facilitating advancements in causal discovery, causal inference, and causal reasoning (Peters et al., 2014). Causal mechanisms are often represented as Structural Causal Models (SCMs), defined as $\mathcal{M} = (U, O, F)$, where $U$ are unobservables, $O$ are observables, and $F$ is a set of structural equations. These equations are expressed as $f_j : X_j = f_j(PA_j, N_j)$, indicating that an outcome variable $F$ depends on its parent variables $PA$ and independent noise $N_j$. Non-linearities in the set of structural equations $F$ influence data complexity and identifiability of causal quantities from observational data (Schölkopf et al., 2012). In an SCM, interventions can be made by setting $X \leftarrow x_1$ and propagating this value through the model $\mathcal{M}$, posing the question of "what *will happen* if I do something?". Counterfactuals expand upon the idea of interventions and are relevant when a value of $X$ is already observed, instead posing the question of "what *would have happened* if something had been different?" In addition to posing a slightly different question, counterfactuals require that exogenous noise terms are held constant, and thus classically require full knowledge of the causal model. In the context of algorithmic fairness, we are limited to level of counterfactuals as protected attributes are typically given and already observed.

In causal reasoning frameworks, one major application of counterfactuals is the estimation of causal effects such as the individual and average treatment effects (ITE and ATE) which quantify the difference and expected difference between outcomes under different values of $X$.

$$ITE : \tau = Y_{X \leftarrow x} - Y_{X \leftarrow x'} \qquad (1)$$

$$ATE : E[\tau] = E[Y_{X \leftarrow x}] - E[Y_{X \leftarrow x'}]. \qquad (2)$$

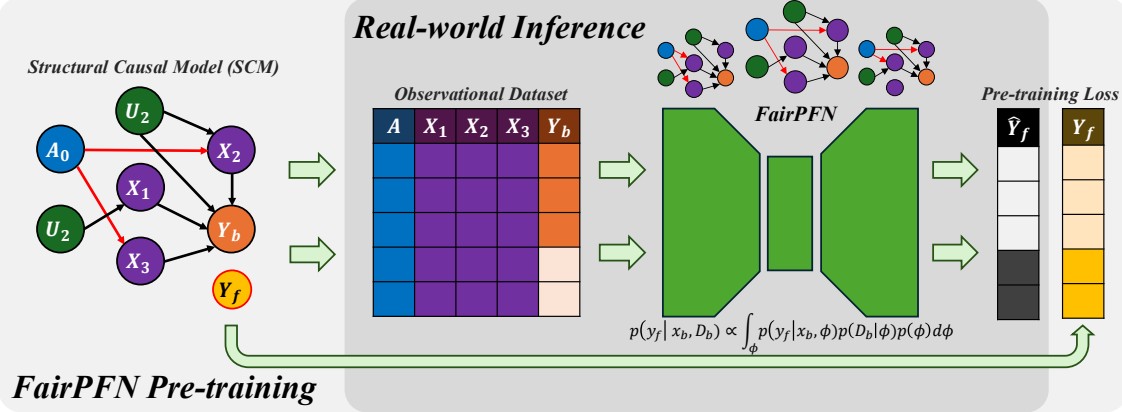

**a) Data generation:** For each pre-training dataset, we generate an SCM and sample a dataset $D$ comprised of a protected attribute $A$, potentially biased observables $X_b$, and biased outcome $Y_b$. We also sample a fair outcome $Y_f$ by removing the outgoing edges of $A$.

**b) Transformer input:** The observational dataset $D$ is partitioned into training and validation splits. Given in-context examples $D_{train}$ the transformer makes predictions on the inference set $D_{val} = (A_{val}, X_{val})$

**c) Fair prediction:** The transformer makes predictions $\hat{Y}_f$ on the validation set, and the pre-training loss is calculated with respect to the fair outcomes in the validation set. The transformer thus learns the mapping $X_b \to Y_f$

*Figure 1.* **FairPFN Overview**: FairPFN is a foundation model for causal fairness, pre-trained on synthetic datasets generated from sparse MLPs that represent SCMs with exogenous protected attributes (a). A biased dataset is created for each MLP/SCM and supplied as context to the transformer (b), with loss computed based on fair outcomes obtained by excluding the causal influence of the protected attribute (c). In practice, (d) FairPFN takes in only an observational dataset to predict fair targets by integrating over the simplest causal explanations for the biased data.

**Counterfactual Fairness** is a foundational notion of causal fairness introduced by Kusner et al. (2017), requiring that an individual's predictive outcome should match that in a counterfactual scenario where they belong to a different demographic group. This notion is formalized in the theorem below.

**Theorem 3.1** (Unit-level/probabilistic). *Given an SCM* $\mathcal{M} = (U, O, F)$ *where* $O = A \cup X$, *a predictor* $\hat{Y}$ *is counterfactually fair on the unit-level if* $\forall \hat{y} \in \hat{Y}, \forall x, a, a' \in A$

$$P(\hat{y}_{A \to a}(u)|X, A = x, a) = P(\hat{y}_{A \to a'}(u)|X, A, = x, a)$$

Kusner et al. (2017) notably choose to model protected attributes as exogenous, which means that they may not be confounded by unobserved variables with respect to outcomes. We note that the definition of counterfactual fairness in Theorem 3.1 is the unit-level probabilistic one as clarified by Plecko & Bareinboim (2024), because counterfactual outcomes are generated deterministically with fixed unobservables $U = u$. Theorem 3.1 can be applied on the dataset level to form the population-level version also provided by Plecko & Bareinboim (2024) which measures the alignment of natural and counterfactual predictive distributions.

**Theorem 3.2** (Population-level). *Given an SCM* $\mathcal{M} = (U, O, F)$ *where* $O = A \cup X$, *a predictor* $\hat{Y}$ *is counterfactually fair on the population-level if* $\forall \hat{y} \in \hat{Y}, \forall x, a, a' \in A$

$$P(\hat{y}_{A \to a}|X, A = x, a) = P(\hat{y}_{A \to a'}|X, A = x, a)$$

Theorem 3.3 can also be transformed into a counterfactual fairness metric by quantifying the difference between natural and counterfactual predictive distributions. In this study we quantify counterfactual fairness as the distribution of the counterfactual absolute error (AE) between predictions in each distribution.

**Definition 3.3** (Absolute Error). Given an SCM $\mathcal{M} = (U, O, F)$ where $O = A \cup X$, the counterfactual *absolute error* of a predictor $\hat{Y}$ is the distribution

$$|P(\hat{y}_{A \to a}(u)|X, A = x, a) - P(\hat{y}_{A \to a'}(u)|X, A = x, a)|$$

We note that because the outcomes are condition on the same noise terms $u$ our definition of AE builds off of Theorem 3.1. Intuitively, when the AE is skewed towards zero, then most individuals receive the same prediction in both the natural and counterfactual scenarios.

Kusner et al. (2017) present various implementations of Counterfactually Fair Prediction (CFP). The three levels of CFP can be achieved by fitting a predictive model $\hat{Y}$ to observable non-descendants if any exist (Level-One), inferred values of an exogenous unobserved variable $K$ (Level-Two), or additive noise terms (Level-Three). Kusner et al. (2017) acknowledge that in practice, Level-One rarely occurs. Level-Two requires that the causal model be *invertible*, which allows the unobservable $K$ to be inferred by abduction. Level-Three models the scenario as an Additive Noise Model, and thus is the strongest in terms of representational capacity, allowing more degrees of freedom than in

Level-Two to represent fair terms. The three levels of CFP are depicted in Appendix Figure 22.

**Causal Fairness** The Causal Fairness Analysis (CFA) framework (Plecko & Bareinboim, 2024) introduces the Standard Fairness Model (SFM), which classifies variables as protected attributes $A$, mediators $X_{med}$, confounders $X_{conf}$, and outcomes $Y$. This framework includes a Fairness Cookbook of causal fairness metrics with a strong analogy to the legal notions of direct and indirect discrimination and business necessity as illustrated in Appendix Figure 23. Plecko & Bareinboim (2024) refute the modeling choice of Kusner et al. (2017) by their inclusion of confounders $X_{conf}$ in the SFM, arguing that these variables contribute to the legally admissible Spurious-Effect (SE).

For simplicity of our experimental results, we follow the modeling of Kusner et al. (2017), and focus on the elimination of the Total-Effect (TE) of protected attributes as defined by Plecko & Bareinboim (2024), while noting in Section 6 the importance of relaxing this assumption in future extensions.

**Prior-data Fitted Networks** Prior-data Fitted Networks (PFNs) (Müller et al., 2022) and TabPFN (Hollmann et al., 2023; 2025) represent a paradigm shift from traditional ML with a causal motivation, namely that simple causal models offer a quality explanation for real-world data. PFNs incorporate prior knowledge into transformer models by pre-training on datasets from a specific prior distribution (Müller et al., 2022). TabPFN, a popular application of PFNs, applies these ideas to small tabular classification tasks by training a transformer on synthetic datasets derived from sparse Structural Causal Models (SCMs). As noted in Hollmann et al. (2023), a key advantage of TabPFN is its link to Bayesian Inference; where the transformer approximates the Posterior Predictive Distribution (PPD), thus achieving state-of-the-art performance by integrating over simple causal explanations for the data.

## 4. Methodology

In this section, we introduce FairPFN, a foundation model for legally or ethically sensitive tabular classification problems that draws inspiration from PFNs and principles of causal fairness. We introduce our pre-training scheme, synthetic data prior, and draw connections to Bayesian Inference to explain the inner workings of FairPFN.

### 4.1. FairPFN Pre-Training

First, we present our pre-training scheme, where FairPFN is fit to a prior of synthetic causal fairness data to identify and remove the causal effects of protected attributes in practice from observational data alone. We provide pseudocode for our pre-training algorithm in Algorithm 2, and outline the

---

**Algorithm 1** FairPFN Pre-training

**Input:**
Number of pre-training epochs $E$ and steps $S$
Transformer $\mathcal{M}$ with weights $\theta$
Hypothesis space of SCMs $\phi \in \Phi$
**begin**
**for** $epoch = 1$ **to** $E$ **do**
    **for** $step = 1$ **to** $S$ **do**
        Draw a random SCM $\phi$ from $\Phi$
        Sample $D_{bias} = (A, X_{bias}, Y_{bias})$ from $\phi$ where A $\{a_0, a_1\}$ is an exogenous binary protected attribute
        Sample $Y_{fair}$ from $\phi$ by performing dropout on outgoing edges of $A$ if any exist
        Partition $D_{bias}$ and $D_{fair}$ into $train/val$
        Pass $D_{bias}^{train}$ into $\mathcal{M}$ as *context*
        Pass $D_{bias}^{val}$ into $\mathcal{M}$ to generate $Y_{pred}^{val}$
        Calculate loss $L = CE(Y_{pred}^{val}, Y_{fair}^{val})$
        Update weights $\theta$ w.r.t $\nabla_\theta L$
    **end for**
**end for**
**Output:** Transformer $\mathcal{M} : X_{bias} \rightarrow Y_{fair}$

---

steps below.

**Data Generating Mechanisms** FairPFN pre-training begins by creating synthetic datasets that capture the causal mechanisms of bias in real-world data. Following the approach of Hollmann et al. (2023), we use Multi-Layer Perceptrons (MLPs) to model Structural Causal Models (SCMs) via the structural equation $f = z(P \cdot W^T x + \epsilon)$, where $W$ denotes activation weights, $\epsilon$ represents Gaussian noise, $P$ is a dropout mask sampled from a log-scale to promote sparsity, and $z$ is a non-linearity. Figure 1 illustrates the connection among sampled MLPs, their corresponding SCMs, and the resulting synthetic pre-training data generated. We note that independent noise terms are not visualized in Figure 1.

**Biased Data Generation** An MLP is randomly sampled and sparsity is induced through dropout on select edges. The protected attribute is defined as a binary exogenous variable $A \in \{a_0, a_1\}$ at the input layer. We uniformly select $m$ features $X$ from the second hidden layer onwards to capture rich representations of exogenous causes. The target variable $Y$ is chosen from the output layer and discretized into a binary variable using a random threshold. A forward pass through the MLP produces a dataset $D_{bias} = (A, X_{bias}, Y_{bias})$ with $n$ samples containing the causal influence of the protected attribute.

**Fair Data Generation** A second forward pass generates a fair dataset $D_{fair}$ by applying dropout to the outgoing edges of the protected attribute $A$ in the MLP, as shown by the red edges in Figure 1. This dropout, similar to that in TabPFN, masks the causal weight of $A$ to zero, effectively reducing its

**Algorithm 2** FairPFN Synthetic Data Generation

  **Input:**
- Number of exogenous causes $U$
- Number of endogenous variables $U \times H$
- Number of features and samples $M \times N$

  **begin**
- Define MLP $\phi$ with depth $H$ and width $U$
- Initialize random weights $W : (U \times U \times H - 1)$
- Sample sparsity masks $P$ with same dimensionality as weights
- Sample $H$ per-layer non-linearities $z_i \sim \{Identity, ReLU, Tanh\}$
- Initialize output matrix $X : (U \times H)$
- Sample location $k$ of protected attribute in $X_0$
- Sample locations of features $X_{biased}$ in $X_{1:H-1}$, and outcome $y_{bias}$ in $X_H$
- Sample protected attribute threshold $a_t$ and binary values $\{a_0, a_1\}$

  **for** $n = 0$ **to** $N$ samples **do**
    - Sample values of exogenous causes $X_0 : (U \times 1)$
    - Sample values of additive noise terms $\epsilon : (U \times H)$
    **for** $i = 0$ **to** $H - 1$ layers **do**
      - Pass intermediate representation through hidden layer $X_{i+1} = z_i(P_i \cdot W_i^T X_i + \epsilon_i)$
    **end for**
    - Select prot. attr. $A$, features $X_{bias}$ and outcome $y_{bias}$ from $X_0$, $X_{1:H-1}$, and $X_H$
    - Binarize $A \in \{a_0, a_1\}$ over threshold $a_t$
    - Set input weights in row $k$ of $W_0$ to 0
    **for** $j = 0$ **to** $H - 1$ layers **do**
      - Pass intermediate representation through hidden layer $X_{j+1} = z_i(P_i \cdot W_j^T X_j + \epsilon_j)$
    **end for**
    - Select the *fair* outcome $y_{fair}$ from $X_H$
  **end for**
- Binarize $y_{fair} \in \{0, 1\}$ and $y_{bias} \in \{0, 1\}$ over randomly sampled output threshold $y_t$
  **Output:** $D_{bias} = (A, X_{bias}, y_{bias})$ and $y_{fair}$

---

influence to Gaussian noise $\epsilon$. This increases the influence of fair exogenous causes $U_0$ and $U_1$ and independent noise terms all over the MLP visualized in Figure 1. We note that $A$ is sampled from an arbitrary distribution $A \in \{a_0, a_1\}$, as opposed to $A \in \{0, 1\}$, since both functions $f = 0 \cdot wx + \epsilon$ and $f = p \cdot 0x + \epsilon$ yield equivalent outcomes. Only after generating the pre-training dataset is $A$ converted to a binary variable for processing by the transformer.

**In-Context Learning** After generating $D_{bias}$ and $D_{fair}$, we partition them into training and validation sets: $D_{bias}^{train}$, $D_{bias}^{val}$, $D_{fair}^{train}$, and $D_{fair}^{val}$. We pass $D_{bias}^{train}$ as context to the transformer to provide information about feature-target relationships. To simulate inference, we input $X_{bias}^{val}$ into the

transformer $\mathcal{M}$, yielding predictions $Y_{pred}$. We then compute the binary-cross-entropy (BCE) loss $L(Y_{pred}, Y_{fair}^{val})$ against the fair outcomes $Y_{fair}^{val}$, which do not contain effects of the protected attribute. Thus, the transformer $\mathcal{M}$ learns the mapping $\mathcal{M} : X_{bias} \to Y_{fair}$.

**Prior-Fitting** The transformer is trained for approximately 3 days on an `RTX-2080` GPU on approximately 1.5 million different synthetic data-generating mechanisms, in which we vary the MLP architecture, the number of features $m$, the sample size $n$, and the non-linearities $z$.

**Real-World Inference** During real-world inference, FairPFN requires no knowledge of causal mechanisms in the data, but instead only takes as input a biased observational dataset and implicitly infers potential causal explanations for the data (Figure 1d) based on the causally generated data it has seen during pre-training. Crucially, FairPFN is provided information regarding which variable is the protected attribute, which is represented in a protected attribute encoder step in the transformer. A key advantage of FairPFN is its alignment with Bayesian Inference, as transformers pretrained in the PFN framework have been shown to approximate the Posterior Predictive Distribution (PPD) (Müller et al., 2022). FairPFN thus approximates a modified PPD, predicting a causally fair target $y_f$ given biased features $X_b$ and a biased dataset $D_b$ by integrating over hypotheses for the SCM $\phi \in \Phi$:

$$p(y_f | x_b, D_b) \propto \int_{\Phi} p(y_f | x_b, \phi) p(D_b | \phi) p(\phi) d\phi \quad (3)$$

This approach has two advantages: it reduces the necessity of precise causal model inference, thereby lowering the risk of *fairwashing* from incorrect models (Ma et al., 2023), and carries with it regularization-related performance improvements observed in Hollmann et al. (2023). We also emphasize that FairPFN is a foundation model and thus does not need to be trained for new fairness problems in practice. Instead, FairPFN performs predictions in a single forward pass of the data through the transformer.

## 5. Experiments

This section assesses FairPFN's performance on synthetic and real-world benchmarks, highlighting its capability to remove the causal influence of protected attributes without user-specified knowledge of the causal model, while maintaining high predictive accuracy.

### 5.1. Baselines

We implement several baselines to compare FairPFN against a diverse set of traditional ML models, causal-fairness frameworks, and fairness-aware ML approaches. We summarize our baselines below, and provide a visualization of

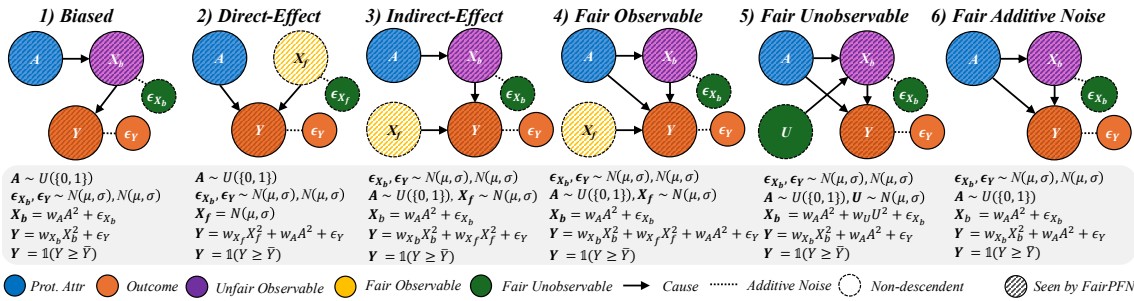

*Figure 2.* **Causal Case Studies:** Visualization and data generating processes of synthetic causal case studies, a handcrafted set of benchmarks designed to evaluate FairPFN's ability to remove various sources of bias in causally generated data. For each group, 100 independent datasets are sampled, varying the number of samples, the standard deviation of noise terms $\sigma$ and the base causal effect $w_A$ of the protected attribute.

our baselines applied to the `Fair Observable` benchmark in Appendix Figure 25.

- `Unfair`: Fit the entire training set $(X, A, Y)$.

- `Unaware`: Fit to the entire training set $(X, A, Y)$. Inference returns the average of predictions on the original test set $(X, A)$ and the test set with alternative protected attribute values $(X, A \rightarrow a')$.

- `Avg. Cnft`: Fit to the entire training set $(X, A, Y)$. Inference returns the average (avg.) of predictions on the original test set $(X, A)$ and the counterfactual (cntf) test set $(X_{A \rightarrow a'}, A \rightarrow a')$.

- `Constant`: Always predicts the majority class

- `Random`: Randomly predicts the target

- `CFP`: Combination of the three-levels of CFP as proposed in Kusner et al. (2017). Fit to non-descendant observables, unobservables, and independent noise terms $(X_{fair}, U_{fair}, \epsilon_{fair}, Y)$.

- `EGR`: Exponentiated Gradient Reduction (EGR) as proposed by (Agarwal et al., 2018) is fit to non-protected attributes $(X, Y)$ with XGBoost (Chen & Guestrin, 2016) as a base model.

In the `CFP`, `Unfair`, `Unaware`, and `Avg. Cntf.` baselines, we employ FairPFN with a random noise term passed as a "protected attribute." We opt to use this UnfairPFN instead of TabPFN so as to not introduce any TabPFN-specific behavioral characteristics or artifacts. We show in Appendix Figure 17 that this reverts FairPFN to a normal tabular classifier with competitive peformance to TabPFN. We also note that our `Unaware` baseline is not the standard approach of dropping the protected attribute. We opt for our own implementation of `Unaware` as it shows improved causal effect removal to the standard approach (Appendix Figure 17).

## 5.2. Causal Case Studies

We first evaluate FairPFN using synthetic causal case studies to establish an experimental setting where the data-generating processes and all causal quantities are known, presenting a series of causal case studies with increasing difficulty to evaluate FairPFN's capacity to remove various sources of bias in causally generated data. The data-generating processes and structural equations are illustrated in Figure 2, following the notation: $A$ for protected attributes, $X_b$ for *biased-observables*, $X_f$ for *fair-observables*, $U$ for *fair-unobservables*, $\epsilon_X$ for *additive noise terms*, and $Y$ for the outcome, discretized as $Y = \mathbb{1}(Y \geq \bar{Y})$. We term a variable $X$ "fair" iff $A \notin anc(X)$. The structural equations in Figure 2 contain exponential non-linearities to ensure the direction of causality is identifiable (Peters et al., 2014), distinguishing the `Fair Unobservable` and `Fair Additive Noise` scenarios, with the former including an unobservable yet identifiable causal effect $U$.

For a robust evaluation, we generate 100 datasets per case study, varying causal weights of protected attributes $w_A$, sample sizes $m \in (100, 10000)$ (sampled on a log-scale), and the standard deviation $\sigma \in (0, 1)$ (log-scale) of additive noise terms. We also create counterfactual versions of each dataset to assess FairPFN and its competitors across multiple causal and counterfactual fairness metrics, such as average treatment effect (ATE) and absolute error (AE) between predictions on observational and counterfactual datasets. We highlight that because our synthetic datasets are created from scratch, the fair causes, additive noise terms, counterfactual datasets, and ATE are ground truth. As a result, our baselines that have access to causal quantities are more precise in our causal case studies than in real-world scenarios where this causal information must be inferred.

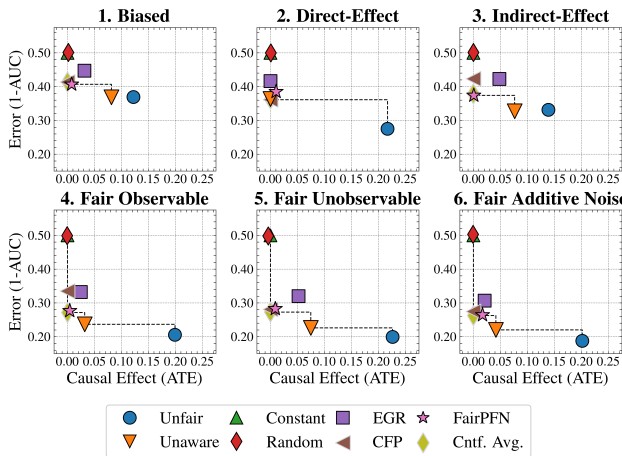

*Figure 3.* **Fairness Accuracy Trade-Off (Synthetic):** Average Treatment Effect (ATE) of predictions, predictive error (1-AUC), and Pareto Front performance of FairPFN versus baselines in our causal case studies. Baselines which have access to causal information are indicated by a light border. FairPFN is on the Pareto Front on 40% of synthetic datasets using only observational data, demonstrating competitive performance with the `CFP` and `Cntf. Avg.` baselines that utilize causal quantities from the true data-generating process.

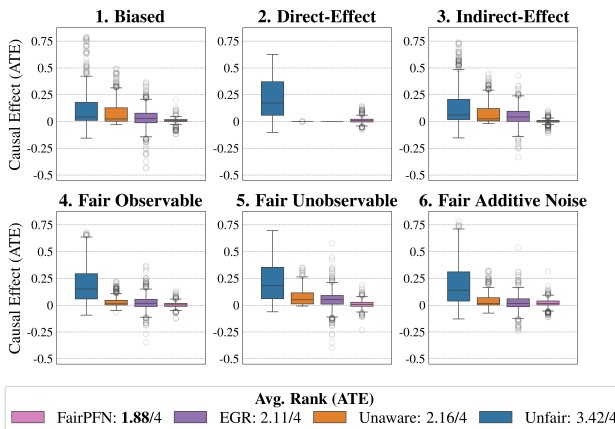

*Figure 4.* **Causal Fairness (Synthetic):** Average Treatment Effect (ATE) of predictions of FairPFN compared to baselines which do not have access to causal information. FairPFN consistently removes the causal effect with a margin of error of (-0.2, 0.2) and achieves an average rank of 1.88 out of 4, only to be outperformed on the `Direct-Effect` benchmark where `Unaware` is the optimal strategy.

**Fairness-Accuracy Trade-Off** Figure 3 presents the fairness-accuracy trade-off for FairPFN and its baselines, displaying the mean absolute treatment effect (ATE) and mean predictive error (1-AUC) observed across synthetic datasets, along with the Pareto Front of non-dominated solutions. FairPFN (which only uses observational data) attains Pareto Optimal performance in 40% of the 600 synthetic datasets, exhibiting a fairness-accuracy trade-

off competitive with `CFP` and `Cntf. Avg.`, which use causal quantities from the true data-generating process. This is even the case in the `Fair Unobservable` and `Fair Additive Noise` benchmark groups, producing causally fair predictions using only observational variables that are either a protected attribute or a causal ancestor of it. This indicates FairPFN's capacity to infer latent unobservables, which we further investigate in Section 5.3. We also highlight how the `Cntf. Avg.` baseline achieves lower error than `CFP`. We believe that this is due to `Cntf. Avg.` having access to both the observational and counterfactual datasets, which implicitly contains causal weights and non-linearities, while `CFP` is given only fair unobservables and must infer this causal information. The fact that a PFN is used as a base model in `Cntf. Avg.` could further explain this performance gain, as access to more observable variables helps guide the PFN toward accurate predictions realistic for the data. We suggest that this `Cntf. Avg.` as an alternative should be explored in future studies.

**Causal Effect Removal** We evaluate FairPFN's efficacy in causal effect removal by analyzing box plots depicting the median, interquartile range (IQR), and average treatment effect (ATE) of predictions, compared to baseline predictive models that also do not access causal information (Figure 4). We observe that FairPFN exhibits a smaller IQR than the state-of-the-art bias mitigation method `EGR`. In an average rank test across 600 synthetic datasets, FairPFN achieves an average rank of 1.88 out of 4. We provide a comparison of FairPFN against all baselines in Figure 24. We note that our case studies crucially fit our prior assumptions about the causal representation of protected attributes. We show in Appendix Figure 13 that FairPFN reverts to a normal classifier when, for example, the exogeneity assumption is violated.

**Ablation Study** We finally conduct an ablation study to evaluate FairPFN's performance in causal effect removal across synthetic datasets with varying size, noise levels, and base rates of causal effect. Results indicate that FairPFN maintains consistent performance across different noise levels and base rates, improving in causal effect removal as dataset size increases and causal effects become easier to distinguish from spurious correlations (Dai et al., 1997). We note that the variance of FairPFN, illustrated by box-plot outliers in Figure 4 that extend to 0.2 and -0.2, is primarily arises from small datasets with fewer than 250 samples (Appendix Figure 11), limiting FairPFN's ability to identify causal mechanisms. We also show in Appendix Figure 14 that FairPFN's fairness behavior remains consistent as graph complexity increases, though accuracy drops do to the combinatorially increasing problem complexity.

For a more in-depth analysis of these results, we refer to Appendix B.

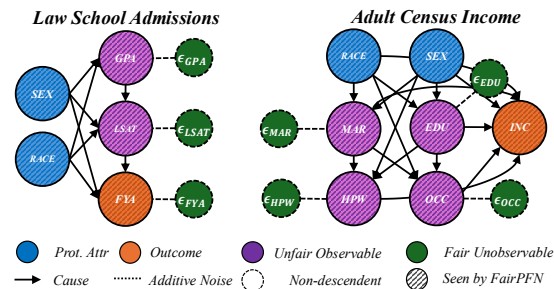

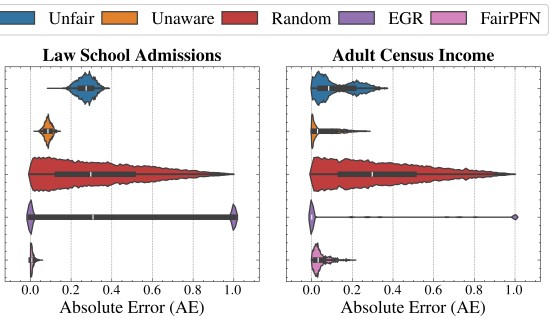

*Figure 5.* **Real-World Datasets**: Assumed causal graphs of real-world datasets Law School Admissions and Adult Census Income.

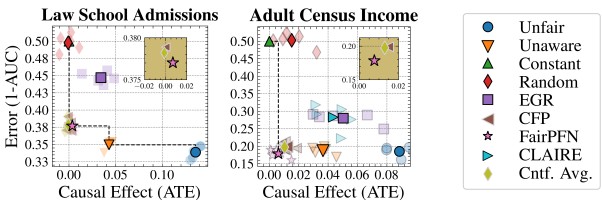

*Figure 6.* **Fairness-Accuracy Trade-off (Real-World):** Average Treatment Effect (ATE) of predictions, predictive error (1-AUC), and Pareto Front of the performance of FairPFN compared to our baselines on each of 5 validation folds (light) and across all five folds (solid) of our real-world datasets. Baselines which have access to causal information have a light border. FairPFN matches the performance of baselines which have access to inferred causal information with only access to observational data.

## 5.3. Real-World Data

This section evaluates FairPFN's causal effect removal, predictive error, and correlation with fair latent variables on two real-world datasets with established causal graphs (Figure 5). For a description of our real-world datasets and the methods we use to obtain causal models, see Appendix A.

**Fairness-Accuracy Trade-Off** We evaluate FairPFN's effectiveness on real-world data in reducing the causal impact of protected attributes while maintaining strong predictive accuracy. Figure 6 shows the mean prediction average treatment effect (ATE) and predictive error (1-AUC) across 5 K-fold cross-validation iterations. FairPFN achieves a prediction ATE below 0.01 on both datasets and maintains accuracy comparable to `Unfair`. Furthermore, FairPFN exhibits lower variability in prediction ATE across folds compared to `EGR`, indicating stable causal effect removal[1].

**Counterfactual Fairness** Next, we evaluate the counterfactual fairness of FairPFN on real-world datasets as introduced in Section 3, noting that the following analysis is conducted at the individual sample level, rather than at the

*Figure 7.* **Counterfactual Fairness (Real-World):** Distributions of Absolute Error (AE) between predictive distributions on observational and counterfactual datasets. Compared to baselines that do not have access to causal information, FairPFN achieves the lowest median and maximum AE on both datasets.

dataset level. Figure 7 illustrates the distribution of Absolute Error (AE) achieved by FairPFN and baselines that do not have access to causal information. FairPFN significantly reduces this error in both datasets, achieving maximum divergences of less than 0.05 on the Law School dataset and 0.2 on the Adult Census Income dataset. For a visual interpretation of the AE on our real-world datasets we refer to Appendix Figure 16.

In contrast, `EGR` performs similarly to `Random` in terms of counterfactual divergence, confirming previous studies which show that optimmizing for group fairness metrics does not optimize for individual level criteria (Robertson et al., 2024). Interestingly, in an evaluation of group fairness metric Statistical Parity (DSP) FairPFN outperforms `EGR` on both our real-world data and causal case studies, a baseline was specifically optimized for this metric (Appendix Figures 20 and 21).

**Trust & Interpretability** In order to build trust in FairPFN and explain its internal workings, we first perform a feature correlation analysis of FairPFN and baseline models using the Law School Admissions dataset. We measure the Kendall rank correlation between observable variables "LSAT" and "UGPA," and inferred noise terms $\epsilon_{LSAT}$ and $\epsilon_{UGPA}$, with predicted admission probabilities $F\hat{Y}A$.

Figure 8 shows that despite only having access to observational data, FairPFN's predictions correlate with fair noise terms similarly to `CFP` which was fit solely to these variables. This result suggests FairPFN's ability to not only integrate over realistic causal explanations for the data, but also correctly remove the causal effect of the protected attribute such that its predictions are influenced only by fair exogenous causes. We note that while FairPFN mitigates the effect of "Race," it increases the correlation of "Sex" compared to the `Unfair` and `CFP` baselines. We discuss how future versions of FairPFN can tackle the problem of *intersectionality* in Section 6. We also further investigate this

---

[1]We note that we also evaluate a pre-trained version of CLAIRE (Ma et al., 2023) on the Adult Census income dataset, but observe little improvement to `EGR`

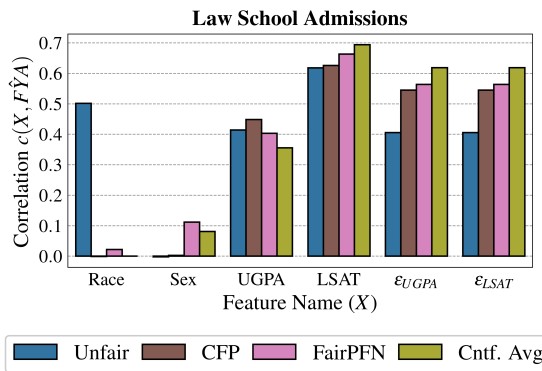

*Figure 8.* **Feature Correlation (Law School):** Kendall Tau rank correlation between feature values and the predictions FairPFN compared to our baseline models. FairPFN produces predictions that correlate with fair noise terms $\epsilon_{UGPA}$ and $\epsilon_{LSAT}$ to a similar extent as the `CFP` baseline, variables which it has never seen in context-or at inference.

result in Appendix Figure 12, which confirms that FairPFN does not remove the effect of additional protected attributes other than the one specified.

We also observe in Figure 3 and 6 the strong performance of our `Cntf. Avg.` baseline, which predicts the average outcome probability in the observational and counterfactual worlds. We thus carry out a similarity test to `Cntf. Avg.` in Appendix Tables 1 and 2, calculating for each other baseline the mean difference in predictions, the standard deviation of this distribution, and the percentage of outliers. We find that FairPFN's predictions are among the closest to this target, with a mean error on synthetic datasets of 0.00±0.06 with 1.87% of samples falling outside of three standard deviations, and a mean error on real-world datasets of 0.02±0.04 with 0.36% of outlying samples.

## 6. Future Work & Discussion

This study introduces FairPFN, a tabular foundation model pretrained to minimize the causal influence of protected attributes in binary classification tasks using solely observational data. FairPFN overcomes a key limitation in causal fairness by eliminating the need for user-supplied knowledge of the true causal graph, facilitating its use in complex, unidentifiable causal scenarios. This approach enhances the applicability of causal fairness and opens new research avenues.

**Extended Problem Scope** We limit our experimental scope to a simple testable setting with a single, binary protected attribute but believe that our prior and transformer architecture can be extended to handle multiple, non-binary protected attributes, addressing both their individual effects and intersectional interactions. We also suggest that FairPFN is capable of predicting not only a fair binary target but also

accommodating multi-objective scenarios (Lin et al., 2019), regression problems (Hollmann et al., 2025), and time series (Hoo et al., 2025). Additionally, FairPFN can generate causally fair versions of previously unfair observables, improving prediction explainability. This enables practitioners to use FairPFN as a fairness preprocessing technique while employing their preferred predictive models in practical applications.

**PFNs for Causal ML** FairPFN implicitly provides evidence for the efficacy of PFNs to perofm causal tasks, and we believe that our methodology can be extended to more complex challenges both within and outside of algorithmic fairness. In algorithmic fairness, one promising extension could be path-specific effect removal (Chiappa, 2019). For example, in medical diagnosis, distinguishing social effects of sex (e.g., sampling bias, male-focus of clinical studies) from biological effects (e.g., symptom differences across sex) is essential for fair and individualized treatment and care. Beyond fairness, we believe PFNs can predict interventional and counterfactual effects, with the latter potentially facilitating FairPFN's evaluation in real-world contexts without relying on estimated causal models. Currently, FairPFN can also mitigate the influence of binary exogenous confounders, such as smoking, on the prediction of treatment success.

**Alignment to Anti-Discrimination Law** Future versions of FairPFN could also relax the assumption of exogenous protected attributes, enabling differentiation between legally admissible spurious effects and direct or indirect effects. Another key concept proposed by (Plecko & Bareinboim, 2024) introduces "Business Necessity" (BN) variables that allow the impact of the protected attribute to indirectly contribute to outcomes to achieve a specified business objectives, such as a research company hiring doctorate holders. In EU law, the analogous "objective justification" concept necessitates a "proportionality test," asserting that justifiable indirect effects must persist only as necessary (Weerts et al., 2023). We contend that proportionality bears a causal interpretation, akin to counterfactual explanations (Wachter et al., 2018).

## Impact Statement

This study attempts to overcome a current limitation in causal fairness, making what we believe is a useful framework for addressing algorithmic discrimination, more accessible to a wider variety of complex fairness problems. While the goal of this work is to have a positive impact on a problem we think is crucial, we acknowledge that we our perspective on fairness is limited in scope to align with EU/US legal doctrines of anti-discrimination. These doctrines are not representative of the world as a whole, and even within these systems, there are vastly different normative viewpoints regarding what constitutes algorithmic fairness and justice.

## Acknowledgements

The authors of this work would like to thank the reviewers, editors and organizers of ICML '25. We would like to additionally thank the Zuse School ELIZA Master's Scholarship Program for their financial and professional support of our main author. We would finally like to thank Sai Prasanna, Magnus Bühler, and Prof. Dr. Thorsten Schmidt for their insights, feedback, and discussion.

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

## A. Real-World Datasets

**Law School Admissions**   The first dataset is the Law School Admissions dataset from the 1998 LSAC National Longitudinal Bar Passage Study (Wightman, 1998), which includes admissions data fr approximately 30,000 US law school applicants, revealing disparities in bar passage rates and first-year averages by ethnicity. We generate counterfactual data and measure causal effects using a slightly different causal model than what was originally proposed by Kusner et al. (2017), which additionally includes edges "UGPA" → "LSAT" and "LSAT" → "FYA". These edges have a plausible temporal explanation, and create a more realistic scenario where "Race" and "Sex" have both a direct and indirect effect on first year averages.

**Causal Modeling with DoWhy**   We use the causal graph in Figure 5 (left) and observational data as inputs for the `dowhy.gcm` module (Sharma & Kiciman, 2020), employing an automated search using the `dowhy.gcm.auto`, which selects the best predictive model in a model zoo of non-linear tree-based models to represent each edge, minimizing either the MSE or negative F1-score depending on the distribution of target following Hoyer et al. (2008) and Peters et al. (2011). We apply each models generate counterfactual datasets, allowing for the estimation of the Average Treatment Effect (ATE) and absolute error (AE). We also use the `compute_noise` function to estimate noise terms $\epsilon_{GPA}$ and $\epsilon_{LSAT}$ for our `CFP` baseline.

**Adult Census Income**   The second dataset, derived from the 1994 US Census, is the Adult Census Income problem (Dua & Graff, 2017), containing demographic and income outcome data ($INC \geq 50K$) for nearly 50,000 individuals[2]. We fit a causal model to assess the Average Treatment Effect (ATE) of the protected attribute $RACE$, generate a counterfactual dataset, and calculate noise term values $\epsilon$.

## B. Ablation Study

To evaluate FairPFN's performance across datasets with varying characteristics, we conduct an ablation study comparing the prediction Average Treatment Effect (ATE) of FairPFN and `Unfair` under different noise levels, base rates of the protected attribute's causal effect, and dataset sizes.

**Base Rate Causal Effect**   We analyze the distributions of prediction ATE from FairPFN and `Unfair` across five quintiles (Q1-Q5) of base ATE (Figure 9). FairPFN's prediction ATE remains stable, while `Unfair`'s prediction ATE increases linearly. In datasets within the Biased, Direct Effect, Level-Two, and Level-Three benchmark groups, where the protected attribute has a high base ATE (Q5), FairPFN exhibits a greater tendency for positive discrimination, resulting in negative prediction ATE values.

**Dataset Noise**   Analyzing dataset noise, indicated by the standard deviation (STD) $\sigma$ of exogenous noise in the structural equations Figure 10 shows that FairPFN retains consistency across varying noise levels. Conversely, `Unfair` exhibits decreased and more peaked distributions of prediction ATE as noise increases from Q1 to Q5, suggests that noise terms may obscure causal effects and diminish their observed impact in the data.

**Dataset Size**   Ablation studies on dataset size (Figure 11) show that FairPFN's prediction ATE displays a tighter distribution with larger datasets, indicating improved performance in causal effect removal. This improvement arises from better identification of causal mechanisms as data availability increases, enabling the transformer to distinguish noise from causal effects.

## C. Future Extensions

In this section we expand upon our discussion of future extensions of FairPFN, in order to encourage the community to build upon and expand our approach.

---

[2]We note that Adult has been heavily criticized in the fairness literature (Ding et al., 2021) due to evidence of sampling bias and an arbitrary chosen income threshold, but elect to include it due to its widely accepted causal model and appearance as a benchmark in other similar studies (Ma et al., 2023)

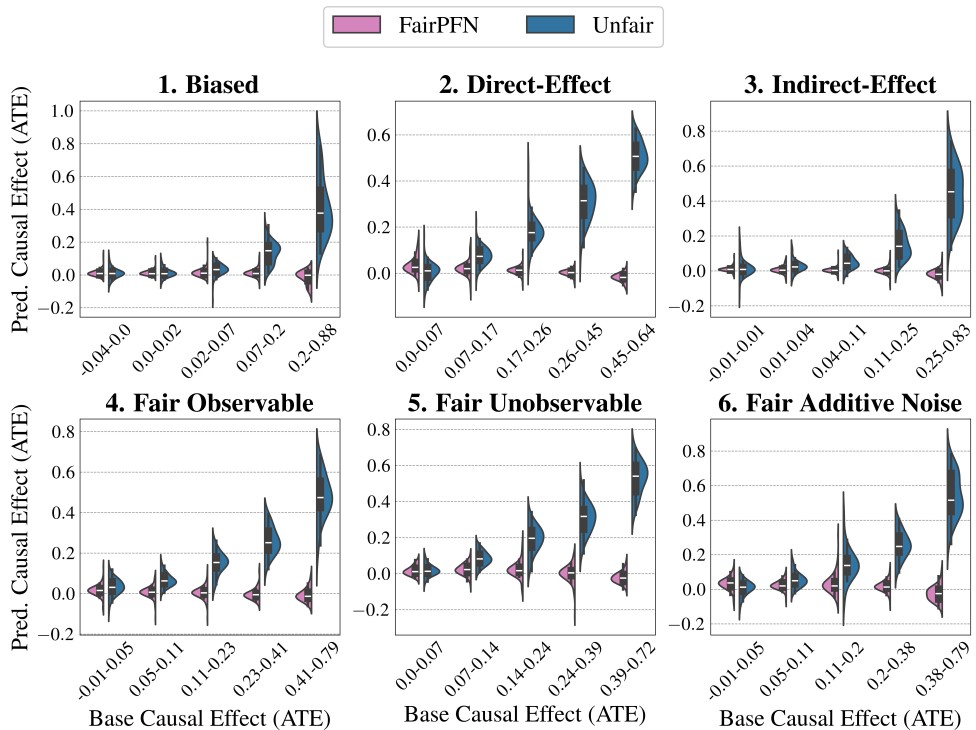

*Figure 9.* **Effect of Base ATE (Synthetic):** Distributions of prediction ATE produced by FairPFN and `Unfair` over quintiles (Q1-Q5) of the protected attributes's base causal effect (base ATE). FairPFN remains consistent across quintiles, sometimes over-correcting and producing a negative prediction ATE in Q5.

**Regression Problems**    FairPFN can be pre-trained as a regression model with very little architectural changes by discretizing continuous output distributions into piecewise intervals and calculating misclassification costs in order to reflect the natural ordering between categories (). Thoroughly evaluated in (Hollmann et al., 2025), such post-proccessing strategies have shown strong performance in tabular regression problems and enable the effective use of classification architectures for continuous targets.

**Protected Attributes in the Wild**    While we limit the scope of this study to binary classification tasks with single, binary protected attributes, we acknowledge that real-world fairness-aware ML problems are often more complex than that. More precisely, protected attributes can be not only binary, but continuous or mulit-category, and discrimination may occur not only with respect to individual protected attributes but with respect to multiple and the interactions between them. Our prior is currently extensible to handle multiple by changing the number of protected attributes that are sampled into each synthetic dataset, removing the outgoing edges of all protected attributes to generate $y_{fair}$, and informing the transformer about which variables are protected attributes. Changing the distribution of protected attributes is also possible, and simply requires transporting the protected attribute into the distribution(s) of choice either before or after its natural continuous value is propagated through the MLP during pre-training.

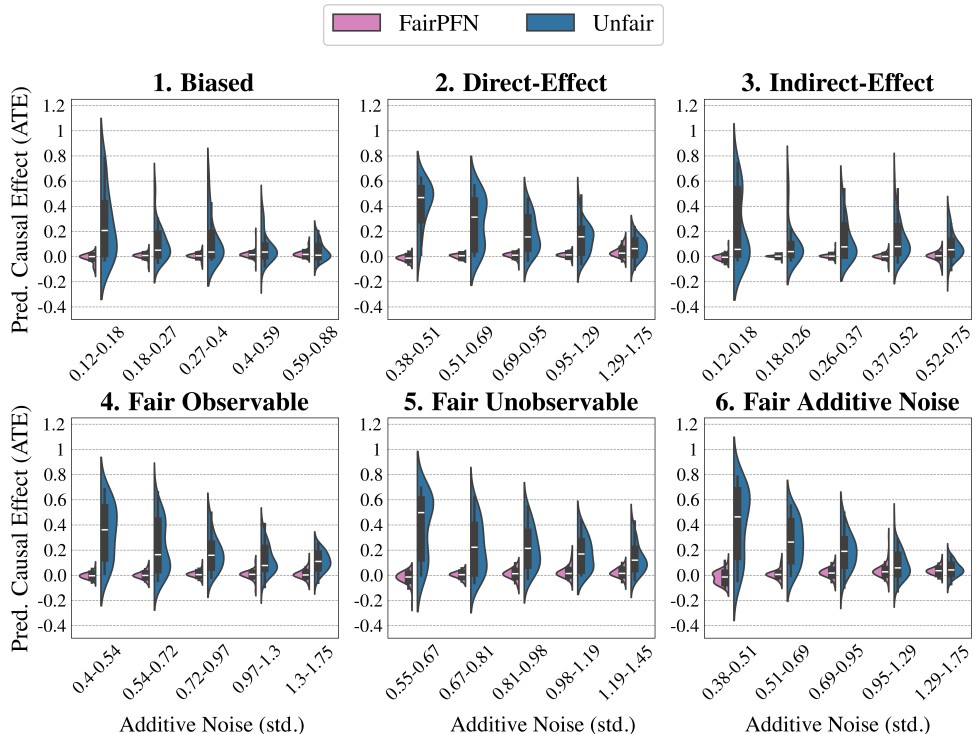

*Figure 10.* **Effect of Dataset Noise (Synthetic):** Distributions of prediction ATE produced by FairPFN and `Unfair` over quintiles (Q1-Q5) of the standard deviation (std.) of exogenous noise terms in the data. FairPFN remains consistent across quintiles, while increased noise decreases the prediction ATE of `Unfair`
.

# D. Supplementary Results

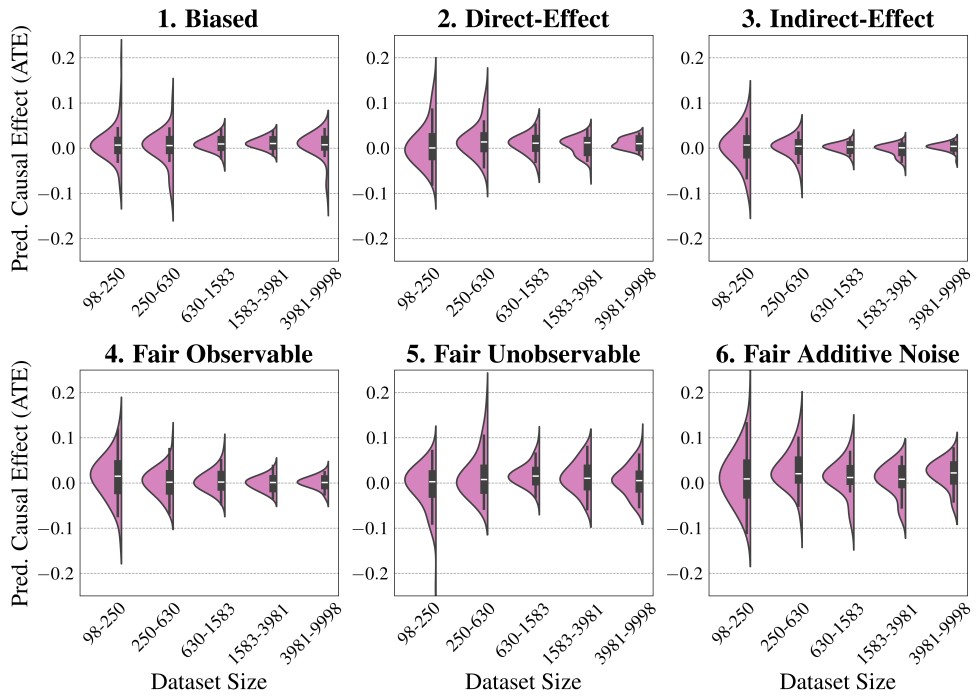

*Figure 11.* **Effect of Dataset Size (Synthetic):** Distributions of prediction ATE produced by FairPFN over quintiles (Q1-Q5) of dataset sizes from 100-10,000 (log-scale). FairPFN becomes better at its task of removing the causal effect of protected attributes when more data is available.

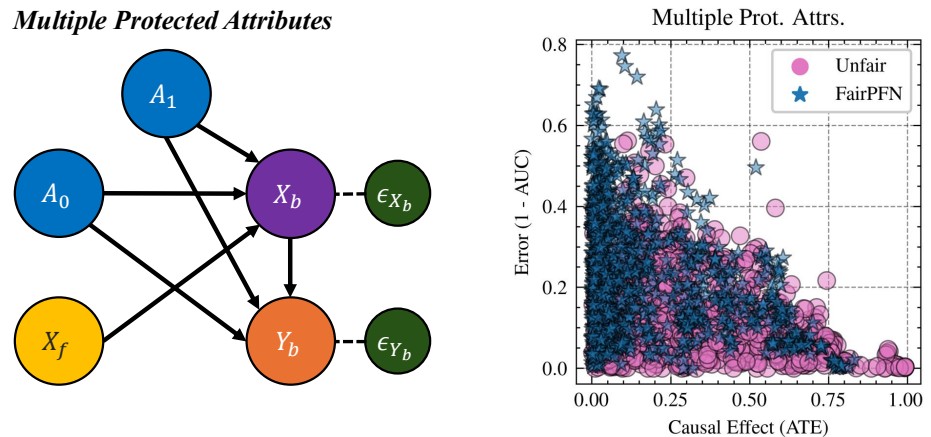

*Figure 12.* **Multiple Protected Attributes (Synthetic):** Distributions of prediction ATE and predictive accuracy produced by FairPFN vs the Unfair predictor when there are multiple protected attributes. This violates FairPFN's prior assumptions and reverts it to a normal classifier.

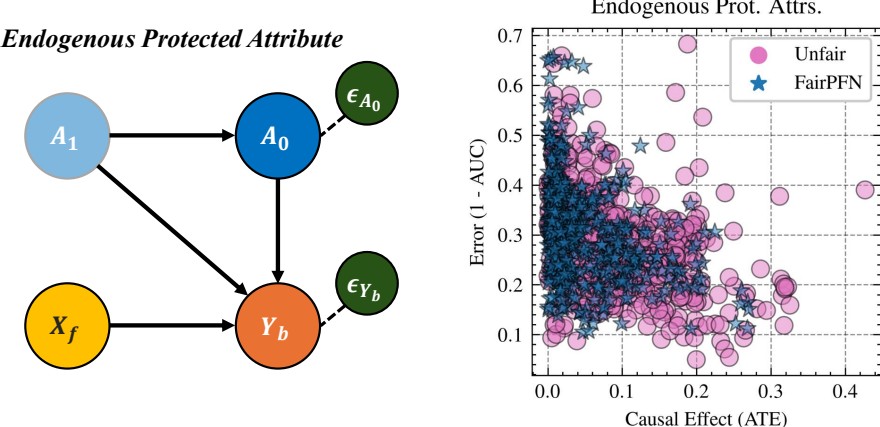

*Figure 13.* **Endogenous Protected Attributes (Synthetic):** Distributions of prediction ATE and predictive accuracy produced by FairPFN vs the Unfair predictor when the protected attribute is *endogenous*. This violates FairPFN's prior assumptions and reverts it to a normal classifier.

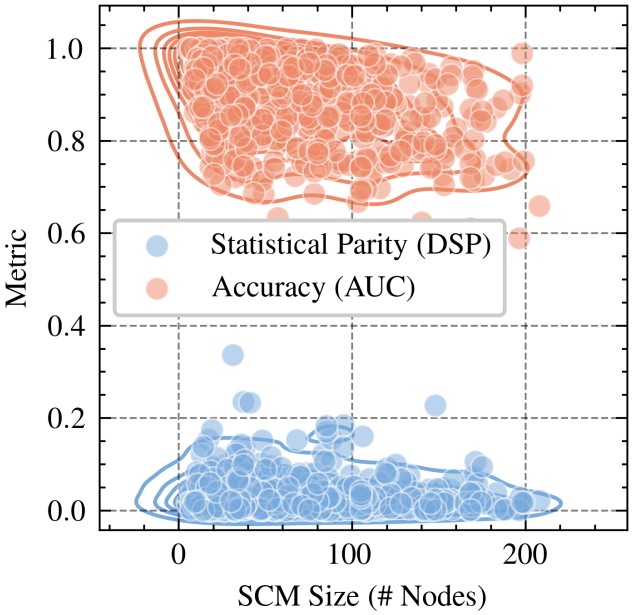

*Figure 14.* **Graph Complexity (Prior):** Distributions of Statistical Parity and predictive accuracy produced by FairPFN on prior samples with graph complexity between 10 and 200 nodes. As graph complexity increases, accuracy drops but fairness remains constant.

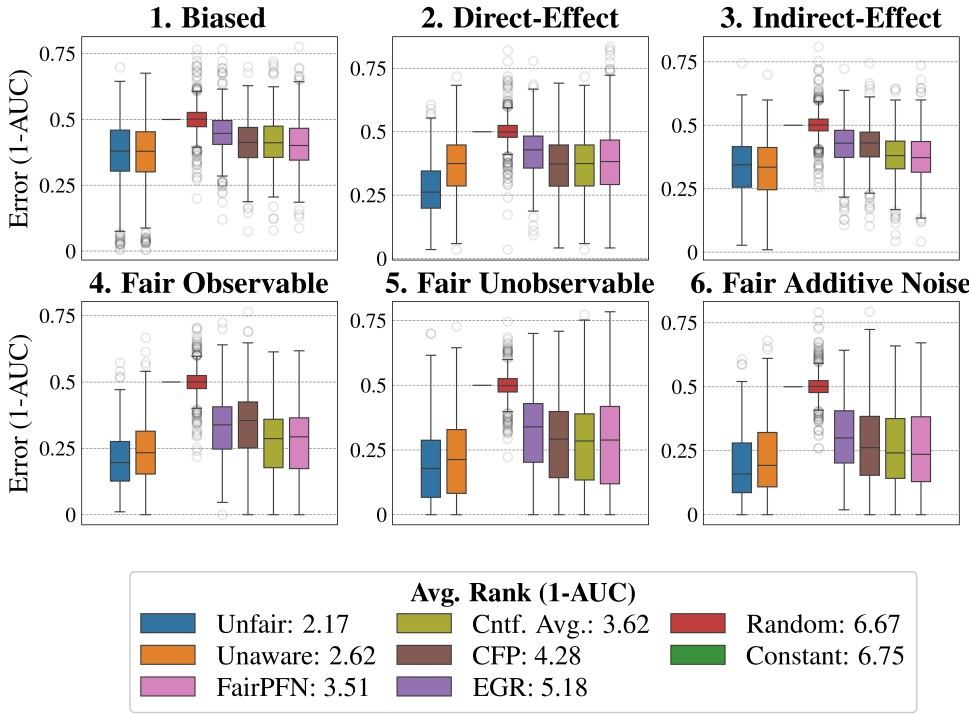

*Figure 15.* **Predictive Error (Synthetic):** Predictive error (1-AUC) of FairPFN compared to our baselines. FairPFN maintains a competitive level of predictive error with traditional ML algorithms, achieving an average rank of 3.51 out of 7.

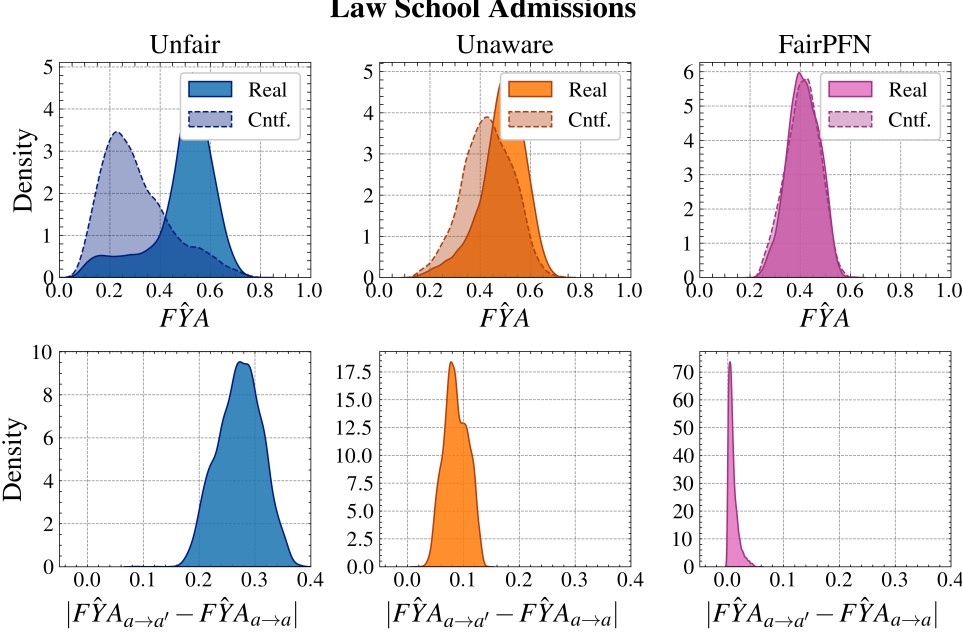

*Figure 16.* **Counterfactual Distributions (Law School):** Predictive distributions of `Unfair`, `Unaware`, and FairPFN on observational and counterfactual versions of the Lawschool Admissions dataset. FairPFN reduces the maximum pairwise difference between these distributions to 0.05.

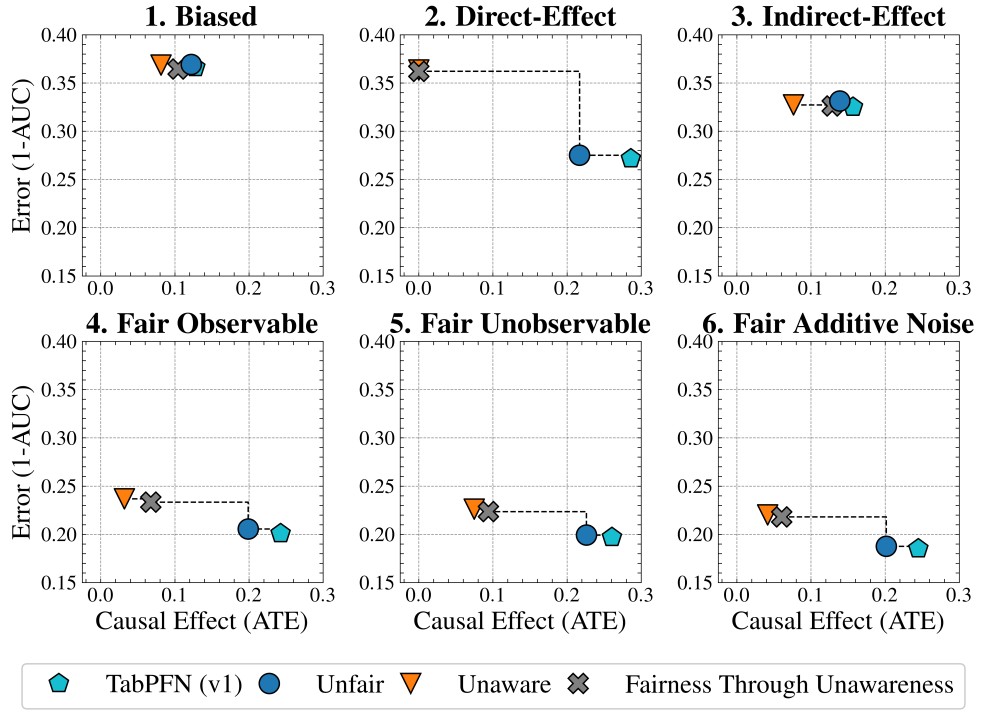

Figure 17. **Baseline Validation (Synthetic)**: Fairness-accuracy trade-off achieved by our baselines `Unfair` and `Unaware` compared to alternative choices of TabPFN (v1) and "Fairness Through Unawareness." `Unfair` achieves competitive performance with TabPFN (v1), while `Unaware` outperforms the standard strategy of dropping the protected attribute from the dataset.

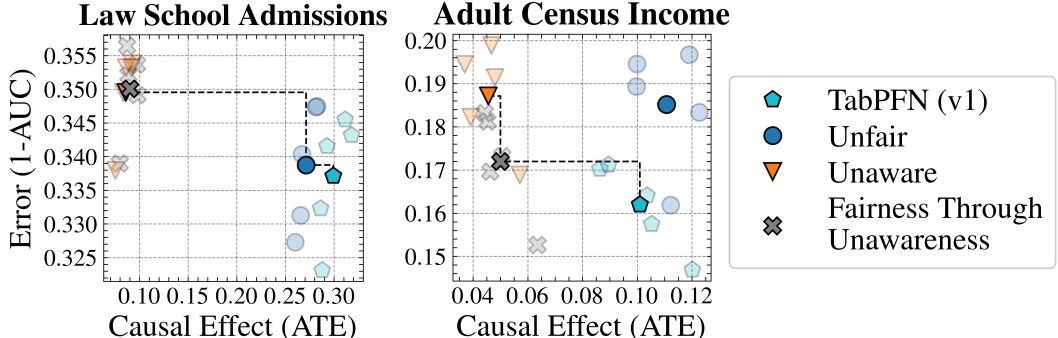

Figure 18. **Baseline Validation (Real-World):** Fairness-accuracy trade-off achieved by our baselines `Unfair` and `Unaware` compared to alternative choices of TabPFN (v1) and "Fairness Through Unawareness." Our choices of baselines achieve competitive performance on the Law School Admissions problem, while alternative baselines perform slightly better on the Adult Census Income problem.

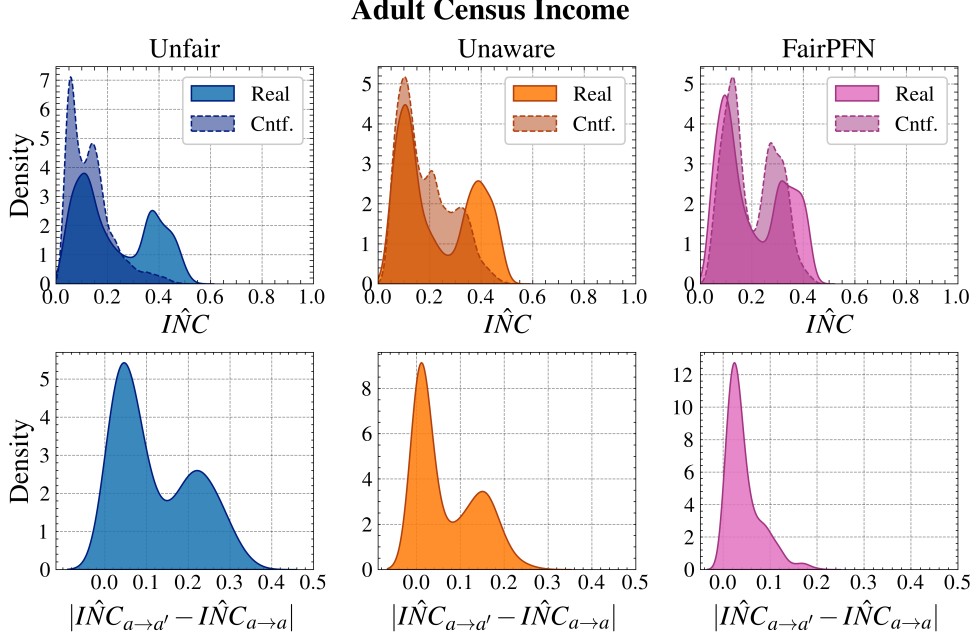

*Figure 19.* **Aligning Counterfactual Distributions (Adult):** Alignment of observational and counterfactual predictive distributions $\hat{Y}$ and $\hat{Y}_{a \to a'}$ on the Adult Census Income problem. FairPFN best aligns the predictive distributions (top) and achieves the lowest mean (0.01) and maximum (0.75) absolute error.

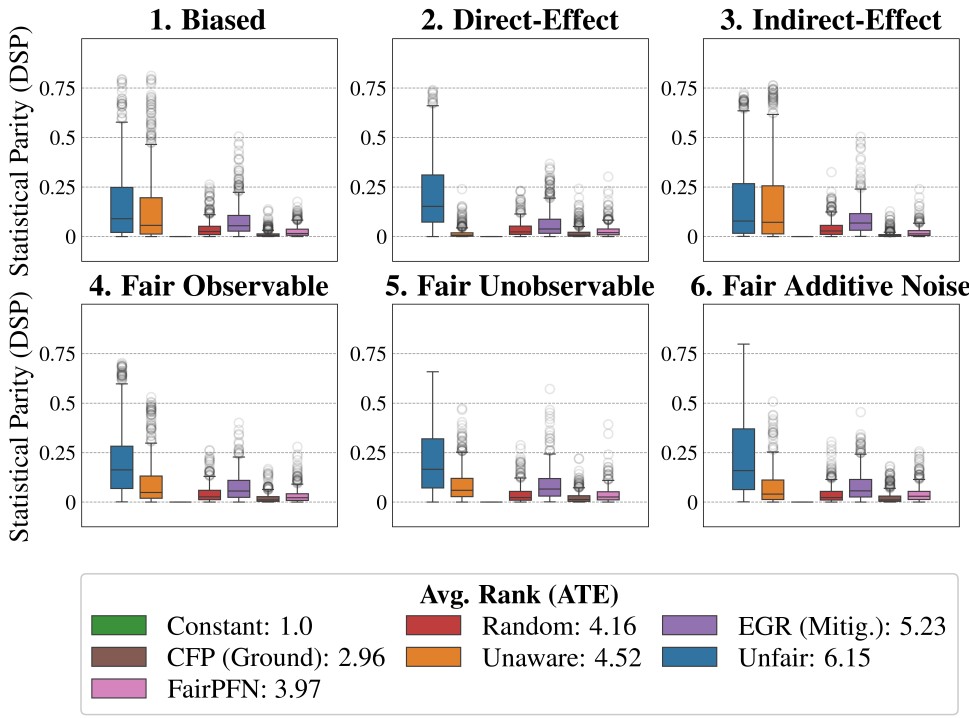

*Figure 20.* **Statistical Parity (Synthetic):** Statistical Parity (DSP) of FairPFN compared to our baselines. FairPFN achieves a similar DSP as the `Random` baseline and outperforms `EGR` which was optimized specifically for this fairness metric, achieving an average rank of 3.97 out of 7.

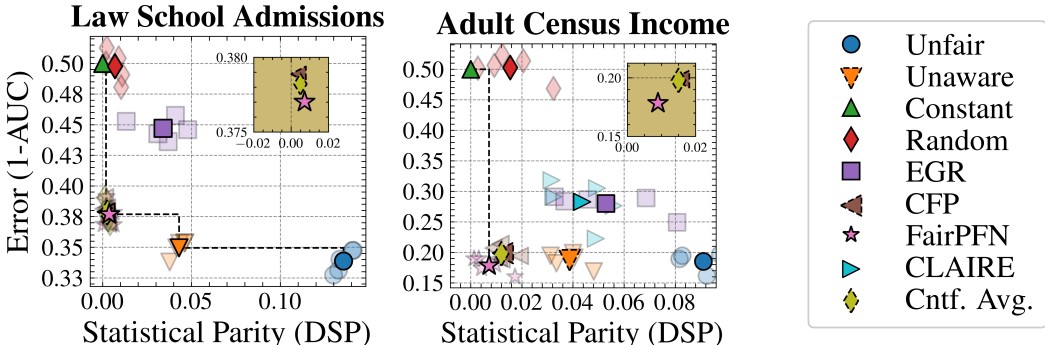

*Figure 21.* **Group-Fairness-Accuracy Trade-off (Real-World):** Statistical Parity (DSP), predictive error (1-AUC), and Pareto Front of the performance of FairPFN compared to our baselines on each of 5 validation folds (light) and across all five folds (solid) of our real-world datasets. FairPFN dominates `EGR` which was specifically optimized for this group fairness metric.

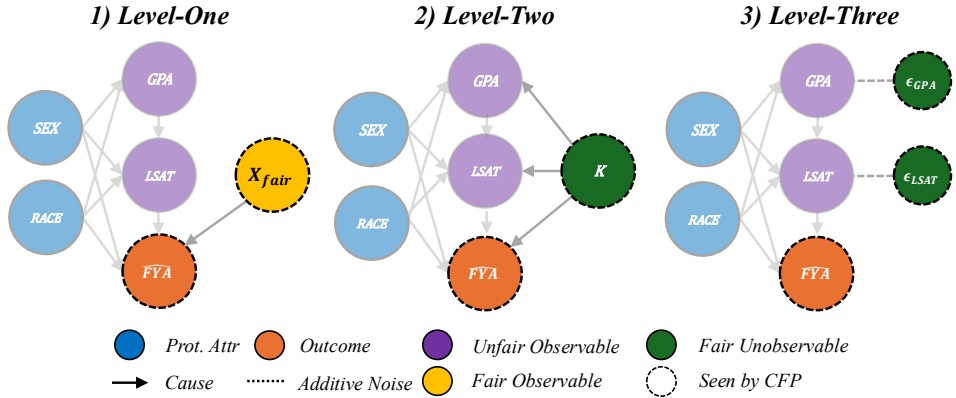

*Figure 22.* **Counterfactually Fair Prediction (CFP):** Three levels of counterfactually fair prediction (CFP) ([Kusner et al., 2017](#)), obtained by fitting a predictor 1) to fair observables (if any exist; left), 2) the inferred values of fair exogenous variables (middle) and 3) the inferred values of independent noise terms (right).

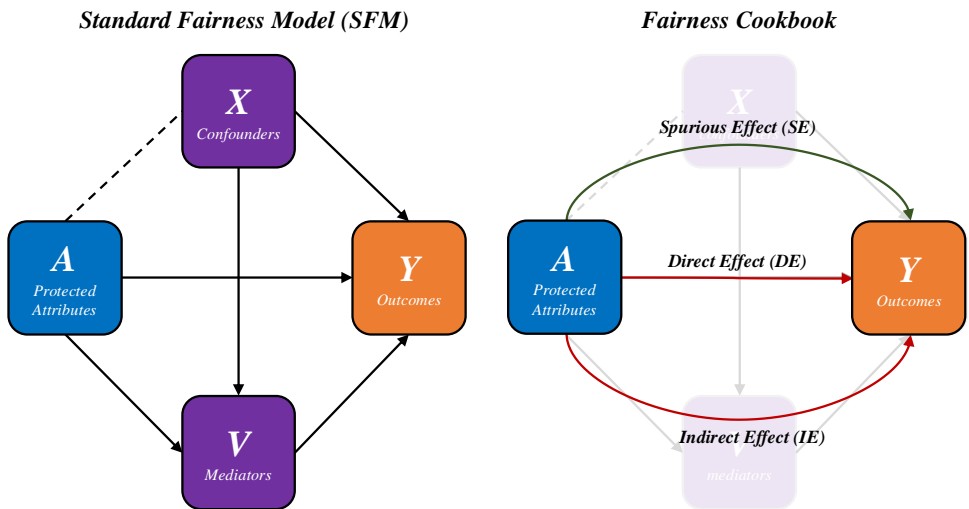

*Figure 23.* **Causal Fairness Analysis (CFA) Framework**: Components of the CFA framework relevant to FairPFN's prior and evaluation. ([Plecko & Bareinboim, 2024](#)) Standard Fairness Model (left; SFM), which provides a meta-model for causal fairness and heavily the design of our prior, and the Fairness Cookbook of causal fairness metrics (right).

| | 1) Biased | 2) Direct-Effect | 3) Indirect-Effect |
|---|---|---|---|
| Unfair | -0.00±0.13 (3.05%) | 0.00±0.14 (0.00%) | -0.00±0.12 (1.65%) |
| Unaware | -0.01±0.09 (2.60%) | 0.00±**0.00** (0.12%) | -0.01±0.08 (1.81%) |
| Constant | -0.36±0.34 (0.00%) | -0.27±0.43 (0.00%) | -0.38±0.34 (0.00%) |
| Random | 0.01±0.30 (0.01%) | 0.01±0.31 (0.01%) | 0.00±0.30 (0.00%) |
| EGR | -0.05±0.46 (0.00%) | -0.07±0.42 (0.00%) | -0.06±0.45 (0.00%) |
| CFP | -0.00±**0.03** (1.31%) | -0.01±**0.03** (0.56%) | -0.01±0.07 (2.29%) |
| FairPFN | 0.00±0.06 (2.03%) | -0.01±**0.03** (1.29%) | -0.00±**0.05** (2.22%) |

| | 4) Fair Observable | 5) Fair Unobservable | 6) Fair Additive Noise | Average |
|---|---|---|---|---|
| Unfair | 0.00±0.14 (0.02%) | -0.00±0.19 (0.00%) | -0.00±0.18 (0.00%) | 0.00±0.15 (0.79%) |
| Unaware | -0.00±**0.05** (2.63%) | -0.00±0.09 (3.68%) | -0.00±0.10 (3.07%) | -0.00±0.07 (2.32%) |
| Constant | -0.49±0.18 (30.10%) | -0.38±0.30 (4.63%) | -0.37±0.33 (0.11%) | -0.38±0.32 (5.81%) |
| Random | 0.01±0.34 (0.00%) | 0.08±0.37 (0.00%) | 0.06±0.37 (0.00%) | 0.03±0.33 (0.00%) |
| EGR | -0.09±0.38 (0.00%) | -0.06±0.39 (0.00%) | -0.07±0.37 (0.00%) | -0.07±0.41 (0.00%) |
| CFP | -0.02±0.14 (1.72%) | 0.00±**0.06** (1.02%) | -0.00±**0.05** (1.00%) | -0.01±**0.06** (1.32%) |
| FairPFN | -0.01±0.07 (1.01%) | 0.01±0.07 (2.20%) | 0.01±0.09 (2.47%) | 0.00±**0.06** (1.87%) |

*Table 1.* **Difference to Cntf. Avg. (Synthetic)**: Mean, standard deviation and percentage of outliers of the predictions on our causal casestudies of FairPFN and our baseline models compared to the predictions of the `Cntf. Avg.` baseline, which shows strong performance in causal effect removal and predictive error due to access to both observational and counterfactual datasets. FairPFN achieves predictions with an average difference to `Cntf. Avg.` of 0.00±0.06, with 1.87% of samples falling outside of three standard deviations.

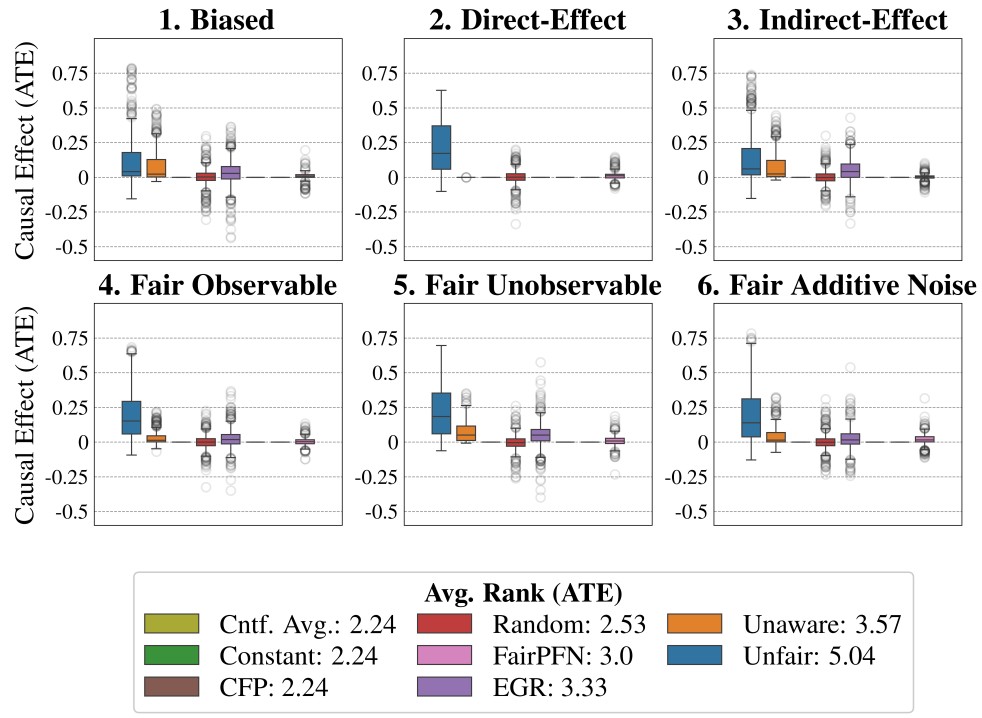

*Figure 24.* **Causal Fairness (Synthetic-All Baselines):** Average Treatment Effect (ATE) of predictions of FairPFN compared to all baselines. FairPFN consistently removes the causal effect with a margin of error of (-0.2, 0.2) and achieves an average rank of 3.0 out of 7.

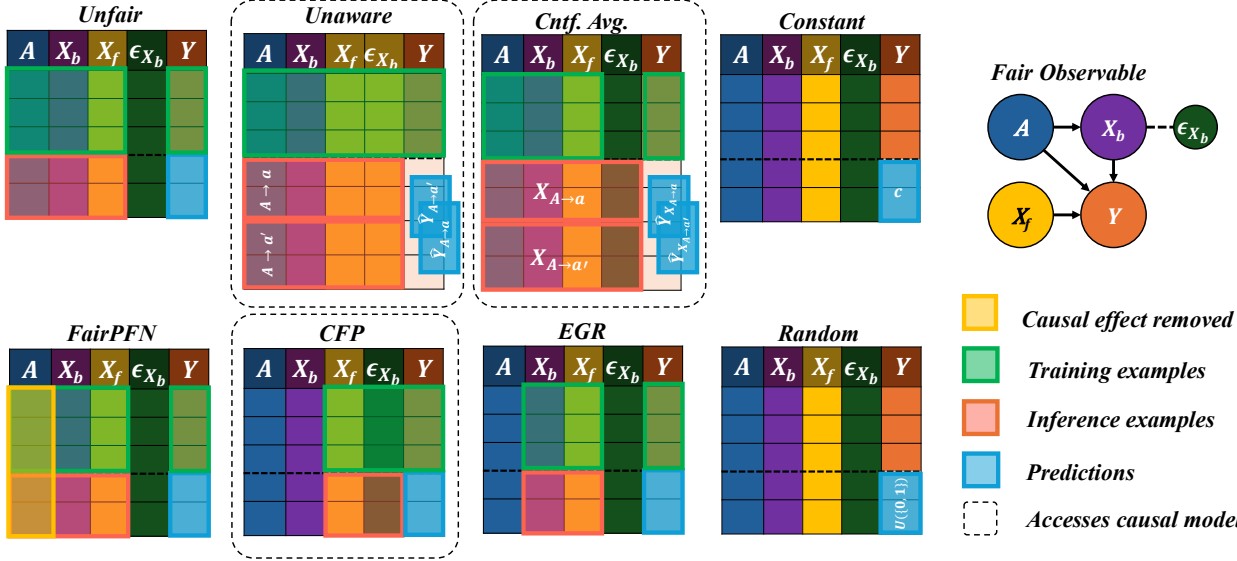

*Figure 25.* **Baseline Models**: Visualization of FairPFN and our baseline models on our `Fair Observable` benchmark group, in terms of which variables each model is fit to and performs inference on on.

| | Law School Admissions | Adult Census Income | Average |
|---|---|---|---|
| Unfair | 0.09±0.10 (0.00%) | 0.05±0.06 (0.60%) | 0.07±0.08 (0.30%) |
| Unaware | 0.03±**0.03** (0.00%) | 0.02±**0.04** (1.49%) | 0.03±**0.04** (0.75%) |
| Constant | -0.40±0.08 (97.51%) | -0.18±0.10 (15.69%) | -0.29±0.09 (56.60%) |
| Random | 0.10±0.30 (0.00%) | 0.32±0.31 (0.30%) | 0.21±0.31 (0.15%) |
| EGR | 0.06±0.45 (0.00%) | 0.01±0.35 (0.00%) | 0.03±0.40 (0.00%) |
| CFP | 0.09±**0.03** (49.21%) | 0.05±0.06 (2.13%) | 0.07±0.05 (25.67%) |
| FairPFN | 0.01±**0.03** (0.11%) | 0.02±**0.04** (0.60%) | 0.02±**0.04** (0.36%) |

*Table 2.* **Difference to Cntf. Avg. (Real)**: Mean, standard deviation and percentage of outliers of the predictions on our real-world datasets of FairPFN and our baseline models compared to the predictions of the `Cntf. Avg.` baseline, which shows strong performance in causal effect removal and predictive error due to access to both observational and counterfactual data. FairPFN achieves predictions with an average difference to `Cntf. Avg.` of 0.02±0.04, with 0.36% of samples falling outside of three standard deviations.

