# OpenReview forum: "FairPFN: A Tabular Foundation Model for Causal Fairness"
_ICML.cc/2025/Conference — ICML 2025 poster_

### Official Review · Reviewer_b4g5 · 2025-03-09

**Overall Recommendation:** 2

**Summary:**

FairPFN is a tabular foundation model designed to address algorithmic bias in machine learning without requiring prior knowledge of the underlying causal model. Existing causal fairness methods rely on predefined causal structures, limiting their applicability in complex real-world scenarios. FairPFN overcomes this by pre-training a transformer on synthetic causal fairness data, where protected attributes are modeled as binary exogenous causes. This foundation model allows FairPFN to identify and mitigate the causal effects of protected attributes using only observational data. The model demonstrates strong performance in both fairness and predictive accuracy across hand-crafted and real-world fairness scenarios, outperforming robust baseline methods. By removing reliance on manually specified causal models, FairPFN makes causal fairness more accessible and provides a scalable solution for mitigating algorithmic discrimination in critical applications.

**Claims And Evidence:**

The paper presents FairPFN as a foundation model for causal fairness, claiming it effectively identifies and removes the causal effects of protected attributes without requiring prior causal knowledge. The empirical results likely support its strong performance in fairness and predictive accuracy across diverse scenarios.

**Essential References Not Discussed:**

Probably not.

**Experimental Designs Or Analyses:**

1. How do we measure counterfactual fairness? The paper uses ATE, but ATE is not equivalent to counterfactual fairness. Feature Correlation can only show the assoication relationship rather than calusa relationship.

2. Causal case studies all focus on sample cases, but what about complex causal structures?

**Methods And Evaluation Criteria:**

Somehow, it makes sense.

**Other Comments Or Suggestions:**

N/A

**Other Strengths And Weaknesses:**

1. This paper is likely an incremental work that applies PFN-related ideas to causal fairness. However, it is well-presented and open-source.

2. The title claims 'Causal Fairness,' while the paper mainly focuses on counterfactual fairness. Please be aware of the distinction between these two definitions.

3. Intervention and counterfactual are two different levels; please be aware and do not mix them up.

**Questions For Authors:**

1. How does the model handle cases where the real-world bias patterns differ from the assumed causal bias model?

2. Can the proposed tabular foundation model work with continuous sensitive attributes? (since it is a tabular foundation model)

**Relation To Broader Scientific Literature:**

This work addresses a key limitation in existing causal fairness frameworks, namely, the requirement for prior knowledge of the causal graph.

**Theoretical Claims:**

1. Theorem 3.1 can be relaxed to form dataset-level counterfactual fairness metrics, such as the Absolute Error (AE) between predictive distributions on real and counterfactual datasets, Equation (2). When the author tries to extend counterfactual fairness to the dataset level, why is the sensitive attribute changed under X rather than Y^hat, as described in Theorem 3.1?

2. [089-091] Counterfactuals evaluate the impact of interventions on outcome variables. Counterfactuals are not the same as interventions. Please specify these two definitions more carefully.

---

> ### Author Rebuttal · Authors · 2025-04-01
>
> We would like to thank you for your detailed response! We have outlined our clarifications and proposed changes below:
>
> >  When the author tries to extend counterfactual fairness to the dataset level, why is the sensitive attribute changed under X rather than Y^hat, as described in Theorem 3.1?
>
> We apologize for the inconsistency in notation. We propose to adopt Plecko et al.'s (2023) notation for unit-level/probabilistic and population-level counterfactual fairness, clarifying that AE is a metric form of their population-level version.
>
> * **Unit-Level/Probablistic:** $P(y_{A\rightarrow a}(u) | X = x, A=a) = P(y_{A\rightarrow a'}(u) | X = x, A=a)$
> * **Population-level:** $P(Y_{A\rightarrow a} | X = x, A=a) = P(Y_{A\rightarrow a'} | X = x, A=a)$
> * **Absolute Error (AE)**  $$AE = |P(Y_{A\rightarrow a} | X = x, A=a) - P(Y_{A\rightarrow a'} | X = x, A=a) |$$
>
> > Counterfactuals evaluate the impact of interventions on outcome variables. Counterfactuals are not the same as interventions.
>
> We appreciate your comment regarding the important distinction between interventions and counterfactuals. We propose to highlight this distinction in the Background section, also noting that counterfactuals crucially hold noise terms constant and thus require full knowledge of the causal model. We will also remove the term "interventional datasets" instead correctly referring to them as counterfactual.
>
> > The title claims 'Causal Fairness,' while the paper mainly focuses on counterfactual fairness.
>
> Thank you for bringing up this question of framing. We understand counterfactual fairness as a specific causal fairness criteria. We believe that this work falls under the umbrella term of "causal fairness" for the following reasons:
>
> 1. Our pre-training objective is to make predictions from a modified SCM with outgoing edges from the protected attributes removed. This deviates from the methodology proposed in the original counterfactual fairness paper (Kusner et. al. 2017)
> 2. One of our evaluation measures is average treatment effect (ATE), which to your point is not a counterfactual fairness measure
> 3. Our key contribution is that causal effect removal can be performed from observational data alone. This contribution is relevant not only to causal fairness, but to the point of other reviewers could be of great impact to causal ML as a whole.
>
> > How do we measure counterfactual fairness? The paper uses ATE, but ATE is not equivalent to counterfactual fairness.
>
> Thank you for bringing up this point of confusion! We define the metric Absolute Error (AE) between predictions on observational and counterfactual datasets to measure counterfactual fairness, not ATE. AE is calculated as follows:
>
> 1. Predict with natural demographics and features
> 2. Predict with counterfactual demographics and counterfactual features
> 3. Take the absolute difference
>
> This process is repeated over the samples in a dataset and is visualized in distribution form as in Figure 7.
>
> > Feature Correlation can only show the association relationship rather than causal relationship.
>
> This is a very valid point. To the best of our knowledge Figure 8 has not made a causal claim. However, we propose explicitly stating that this result should not be interpreted causally.
>
> > Causal case studies all focus on simple cases, but what about complex causal structures?
>
> This comment poses an interesting research question! In order to explore FairPFN's performance on more complex causal structures, we have sampled increasingly complex SCMs (up to 200 nodes) from our prior, and evaluated our model's performance. In the figure linked below, we observe that as SCM size increases, accuracy drops while fairness remains relatively constant. This is an interesting insight that we would include in the appendix.
>
> https://anonymous.4open.science/r/FairPFN-supplementary/figures/complexity.png
>
> > How does the model handle cases where the real-world bias patterns differ from the assumed causal bias model?
>
> This is an important question! We have created a new synthetic case study "Endogenous Protected Attribute" showing that when protected attributes are confounded by unobserved variables, FairPFN reverts to an unfair classifier.
>
> https://anonymous.4open.science/r/FairPFN-supplementary/figures/endogenous.pdf
>
> We also refer to our response to reviewer GboP regarding an additional case study on intersectionality.
>
> > Can the proposed tabular foundation model work with continuous sensitive attributes?
>
> Currently FairPFN handles binary protected attributes, but can be extended to continuous attributes by not binarizing values of $A$ during pre-training. We propose to including this in an appendix section "Future Extensions"
>
> **References**
> 1. Plecko, D et. al. Causal fairness analysis. Foundations and Trends in Machine Learning, 17:304–589, 2024, https://arxiv.org/abs/2207.11385
> 2. Kusner, M., et. al. Counterfactual fairness. NeurIPS’17 pp. 4069–4079, 2017. https://arxiv.org/abs/1703.06856

---

### Official Review · Reviewer_1GNH · 2025-03-10

**Overall Recommendation:** 4

**Summary:**

Let $A$ denote (binary) protected attributes, $X$ features, and $Y$ a binary response variable.  A FairPFN is a transformer trained on synthetic data in such a way that when conditioned on $(A, X_{bias}, Y_{bias})$ it is encouraged to complete a query from the same distribution, denoted $X_{bias}^{val}$ with not $Y_{bias}^{val}$ but $Y_{fair}^{val}$. After extensive pretraining, the FairPFN can be used to predict in a casually fair manner. Specifically, given a new dataset $\mathcal D$, we pass it to the FairPFN as context and then complete new queries in an ICL manner without any updates to the FairPFN.

## update after rebuttal

I maintain my score and positive impression of the paper.

**Claims And Evidence:**

I am uncomfortable with presenting the FairPFN as learning some sort of Bayesian posterior predictive distribution. I think this would be true if the pretraining loss employed was $L(Y_{pred}, Y_{bias}^{val})$. However the actual pretraining loss employed is $L(Y_{pred}, Y_{fair}^{val})$.

In Line 238, “FairPFN thus approximates a modified PPD,...,” pointing to the PPD in Equation 3. This is not mathematically true, or at least I fail to see how this could be true. The pretraining of PFNs no longer targets the underlying Bayes PPD when the query undergoes a distribution shift relative to the context.

**Essential References Not Discussed:**

I think all relevant references are cited and discussed.

**Experimental Designs Or Analyses:**

The experimental design and analyses are sound. A wide array of synthetic and real world data sets are considered. The evaluation metrics are sensible and align with standard ones studied in the fairness literature.

**Methods And Evaluation Criteria:**

The method here consists of synthetic data generation followed by FairPFN pre-training on the synthetic data. This part is solid.

The baselines, synthetic and real datasets considered in the Experiments also seem sound to me.

**Other Comments Or Suggestions:**

* Figure 1 is visually appealing, but I’d still prefer to understand the data generating mechanism through formal equations which is currently missing in the early parts of Section 4.1
* Might be useful to define $A$ in Algorithm 1

**Other Strengths And Weaknesses:**

I really enjoyed reading this paper. I think it’s a big step forward for causal fairness research. I can envision many future papers inspired by this work, for instance where we use other notions of causal fairness to generate synthetic data.

The main weakness that I see is related, again, to the claims that FairPFN is approximating a “modified” Bayes posterior predictive density (PPD). If it is, what is the likelihood-prior pair that underlies said modified Bayes PPD? I don’t think this question needs to necessarily be answered in this paper, but the paper certainly should avoid any chance of being misleading on this matter.

**Questions For Authors:**

1. Would you consider writing a few comments on extending the current methodology to regression settings, i.e., when $Y$ is real-valued. Should we be considered that PFNs do not handle real-valued responses in a very natural way?

**Relation To Broader Scientific Literature:**

This work borrows existing notions of causal fairness and really pushes what can be done by leveraging the power of Transformers. Notably, no algorithm for learning the underlying DAG is needed, allowing us to bypass a key challenge in causal fairness.

**Theoretical Claims:**

The only theoretical claim is that the FairPFN is approximating the PPD in (3) which I have doubts about, see comment under Claims and Evidence.

---

> ### Author Rebuttal · Authors · 2025-04-01
>
> Thank you for your insightful comments! Detailed responses below. Most importantly, we would like to clarify our perspective on the mathematical foundation of our method and how it relates to PPDs.
>
> > In Line 238, “FairPFN thus approximates a modified PPD,...,” pointing to the PPD in Equation 3. This is not mathematically true, or at least I fail to see how this could be true..
>
> > I don’t think this question needs to necessarily be answered in this paper, but the paper certainly should avoid any chance of being misleading on [whether it approximates a modified PPD]
>
> We believe FairPFN indeed approximates the statement from Eq. (3), that is
>
> $$p(y_f|x_b,D_b) \propto \int_{\Phi} p(y_f | x_b, \phi)p(D_b | \phi)p(\phi)d\phi.$$
>
> Here, $\phi$ is the latent, which in our case is the whole SCM, its weights, noise terms, and activations. $\phi$ is sampled per dataset during training. We would like to clarify that the fair outcomes $y_f$ come from the same data generating process $\phi$ as $y_b$ with a deterministic mapping applied to the original SCM, namely the removal of outgoing edges from the protected attribute, with everything else remaining constant. In this way we believe that both SCMs, the original and intervened upon one, are encapsulated in $\phi$ as the distribution shift is deterministic.
>
> We would also be happy to introduce an additional term $\phi'$ and a deterministic mapping from $\phi$ in order to reflect the difference between the original and intervened upon SCM. If your criticism is about calling this a "modified PPD", we are open to recommendations on the exact wording.
>
> We now outline why we believe that FairPFN approximates $p(y_f|x_b,D_b)$ (Eq. 3):
>
> During training we sample examples for our training set and our test set using the likelihood $p(x_b,y_b,y_f|\phi)$, as detailed in the Algorithm at https://anonymous.4open.science/r/FairPFN-supplementary/figures/data_generation.pdf
> We can use this distribution to sample our biased conditioning set $D_b=\{(x^i_b,y^i_b) \sim p(x_b,y_b|\phi)\}_{1 \leq i \leq n}$, as well as our hold-out examples with fair labels.
>
> Our training loss is defined as:
>
> $$E_{\phi \sim p(\phi); (x_b,y_f) \sim p(x_b,y_f|\phi); D_b \sim p(D_b|\phi)}[-log(q_\theta(y_f|x_b,D_b))],$$
>
> where we first sample our latent $\phi$ and then conditioned on it both our test input-output pair and our conditioning set.
>
> We can reformulate this loss as a KL-divergence with the "modified PPD" from above, similar to the derivation by Müller et al. (2022), as follows
> \begin{align}
> &= E_{x_b,y_f,D_b \sim p(x_b,y_f,D_b)}[-log(q_\theta(y_f|x_b,D_b))]\\\\
> &= - \int\int\int p(x_b,y_f,D_b)log(q_\theta(y_f|x_b,D_b))dx_b dy_f dD_b\\\\
> &= - \int\int p(x_b,D_b) \int p(y_f|x_b,D_b) log(q_\theta(y_f|x_b,D_b))dy_f dx_b dD_b\\\\
> &= \int\int p(x_b,D_b) \left[ \text{KL}(p(y_f|x_b,D_b)|| q_\theta(y_f|x_b,D_b)) + \int H(p(y_f|x_b,D_b))\right] dx_b dD_b\\\\
> &= E_{x_b,D_b \sim p(x_b,D_b)}[\text{KL}(p(y_f|x_b,D_b)|| q_\theta(y_f|x_b,D_b))] + C,
> \end{align}
>
> where $H$ is entropy, $KL$ is KL divergence and $C$ is a constant that does not depend on $\theta$.
>
> So minimizing the train loss is the same as minimizing the average KL div. to $p(y_f|x_b,D_b)$, our model $q_\theta$ thus approximates it. Do you agree with this view? We propose to add this derivation to the paper.
>
> > Figure 1 is visually appealing, but I’d still prefer to understand the data generating mechanism through formal equations which is currently missing in the early parts of Section 4.1
>
> We propose to provide an Algorithm 2 in Section 4.1, namely https://anonymous.4open.science/r/FairPFN-supplementary/figures/data_generation.pdf, detailing how synthetic datasets are sampled from our MLP implementation of Structural Causal Models (SCMs).
> Further, our synthetic datasets can be sampled and visualized with https://anonymous.4open.science/r/FairPFN/prior_data_example.ipynb
>
> > Might be useful to define A in Algorithm 1
>
> We will update line 173 to
>
> *Sample $D_{bias} = (A, X_{bias}, Y_{bias})$ from $\Phi$*
>
> and define the terms very explicitly in our new Algorithm 2
>
> > Would you consider writing a few comments on extending the current methodology to regression settings, i.e., when Y is real-valued. Should we be considered that PFNs do not handle real-valued responses in a very natural way?
>
> The recently released version of the TabPFN (Hollmann et al. 2025), whose predecessor we are building upon, integrates regression and one could follow their simple, but, according to their benchmarks, powerful setup of discretizing the continuous space. We will further add an Appendix, where we detail more future work possibilities like this.
>
> **References**
>
> 1. Müller, S., et al. F. Transformers Can Do Bayesian Inference. ICLR, 2022. https://openreview.net/forum?id=KSugKcbNf9
>
> 2. Hollmann, N., et al. Accurate predictions on small data with a tabular foundation model. Nature, 637(8045):319–326, 2025. https://www.nature.com/articles/s41586-024-08328-6

---

### Official Review · Reviewer_GboP · 2025-03-14

**Overall Recommendation:** 3

**Summary:**

The paper introduces FairPFN, a tabular foundation model for causal fairness in machine learning. Pre-trained on synthetic causal fairness data, it mitigates the influence of protected attributes without requiring prior causal knowledge. Experiments show FairPFN effectively removes causal bias while maintaining strong predictive accuracy. Key contributions include a novel pre-training strategy, a synthetic causal data prior, and a fairness-aware foundation model.

## Update after rebuttal

Thanks for author's rebuttal, it solved some of my problems, I'll maintain my rating.

**Claims And Evidence:**

The claims made in the submission are supported by clear and convincing evidence, quantitative results showing improvements in causal fairness metrics (ATE) and predictive accuracy (1-AUC) across synthetic and real-world datasets. Meanwhile visual comparisons and ablation studies demonstrating the effectiveness of FairPFN in removing causal effects.

**Essential References Not Discussed:**

N/A

**Experimental Designs Or Analyses:**

The author use a diverse set of synthetic causal case studies with varying complexity and known ground truth. Also evaluate on real-world datasets with established causal graphs. Additionally, they compare against multiple relevant baselines, including both traditional ML models and causal fairness methods.

**Methods And Evaluation Criteria:**

The proposed methods and evaluation criteria are well-suited for causal fairness in tabular data. Synthetic causal case studies enable controlled experiments with known ground truth, ensuring rigorous testing. Standard metrics like ATE and predictive error assess both fairness and accuracy. Comparisons with multiple baselines, from traditional ML models to causal fairness frameworks, further validate FairPFN’s effectiveness.

**Other Comments Or Suggestions:**

The authors might consider releasing the code and pre-trained models to facilitate reproducibility and further research in this area.

Some figures (e.g., Figure 1) are quite complex and might benefit from additional explanation or simplification for better understanding.

**Other Strengths And Weaknesses:**

The model is tested on real-world datasets with established causal graphs, demonstrating its practical utility. Meanwhile, The paper is well-structured and clearly explains the methodology, experiments, and results, the figures and tables effectively support the text.

However, the pre-training process requires significant computational resources (3 days on an RTX-2080 GPU), which might limit accessibility for some researchers. Meanwhile, the paper notes that FairPFN increases the correlation of "Sex" with predictions in one analysis, suggesting potential issues with intersectional fairness that need further investigation.

**Questions For Authors:**

See "Other Comments Or Suggestions" and "Other Strengths And Weaknesses".

**Relation To Broader Scientific Literature:**

FairPFN addresses limitations in existing causal fairness frameworks that require prior knowledge of causal models. It builds on concepts from counterfactual fairness and the Causal Fairness Analysis (CFA) framework, while relaxing assumptions about the need for user-specified causal information. Also FairPFN extends the PFN paradigm to causal fairness, leveraging the success of models like TabPFN in small tabular classification tasks. It demonstrates how PFNs can be adapted for complex causal tasks, opening new research avenues in causal ML.

**Theoretical Claims:**

The theoretical claims about FairPFN's ability to approximate the Posterior Predictive Distribution (PPD) and integrate over causal explanations are plausible and align with previous work on Prior-data Fitted Networks (PFNs). The connection to Bayesian Inference is well-established, and the modification to focus on causally fair targets is logically consistent with the goals of the paper. No specific proofs are provided, but the conceptual framework is sound and builds on existing theoretical foundations in causal ML and PFNs.

---

> ### Author Rebuttal · Authors · 2025-04-01
>
> We thank you for your constructive feedback! In response, we have: (1) clarified that FairPFN requires no retraining for new applications (2) added analysis on intersectional fairness through a synthetic case study; (3) clarified the availability of inference and pre-training data generation code; and (4) simplified Figure 1 while adding a supporting pseudocode algorithm detailing the pre-training data generation process. We welcome any additional guidance should you have further suggestions to enhance our work.
>
> > However, the pre-training process requires significant computational resources (3 days on an RTX-2080 GPU), which might limit accessibility for some researchers.
>
> We apologize for the confusion! In fact, FairPFN is a pre-trained foundation model that requires no retraining for researchers wishing to reproduce our results or apply it to new fairness problems. We propose adding a sentence at the end of Section 4.1 (Real World Inference) stating: "As a pre-trained foundation model, FairPFN can be directly applied to new fairness problems through a single forward pass of data through the transformer, eliminating any need for retraining." Our inference and synthetic data code is available at https://anonymous.4open.science/r/FairPFN
>
> > FairPFN increases the correlation of "Sex" with predictions in one analysis, suggesting potential issues with intersectional fairness that need further investigation.
>
> We appreciate your concern regarding the increased correlation of “Sex” in Figure 8. The reason for this is that FairPFN is currently pre-trained for binary classification tasks with single, binary protected attributes, so it is simply not tasked to mitigate the effects of secondary protected attributes.
>
> In order to further investigate this result, we have created a synthetic case study "Multiple Protected Attributes" demonstrating that while FairPFN removes the effect of the first protected attribute, the causal effect of secondary protected attributes resembles that of a non-fairness aware classifier. A visualization of our new "Multiple Protected Attributes" case study and an illustration of this result is linked below:
>
> https://anonymous.4open.science/r/FairPFN-supplementary/figures/multiple.pdf
>
> We propose the following changes:
> 1. Highlight this limitation in the results section
> 2. Include the causal graph visualization of this new case study in the appendix, as well as the above figure illustrating how FairPFN reverts to a non-fairness aware classifier regarding secondary protected attributes
> 3. Detail methodological changes in an appendix section titled “Future Extensions” for addressing the challenge of intersectionality.
>     - This would include sampling multiple protected attributes as exogenous causes in the synthetic pre-training data generation and informing the transformer via an encoding mask which of these multiple variables is protected. Then in order to generate the fair outcomes, dropout would need to be performed on the outgoing edges of all simulated protected attributes.
>
> > The authors might consider releasing the code and pre-trained models to facilitate reproducibility and further research in this area.
>
> We appreciate your emphasis on reproducibility! We have actually already provided these resources in our initial submission, specifically an inference pipeline to access our pre-trained model and the code to generate our synthetic pre-training data. Our Anonymous GitHub Repository link is included at the end of Section 1: https://anonymous.4open.science/r/FairPFN
>
> > Some figures (e.g., Figure 1) are quite complex and might benefit from additional explanation or simplification for better understanding.
>
> Thank you for this feedback regarding the clarity of our figures! We've modified Figure 1 to focus exclusively on visualizing the SCM from which our synthetic pre-training data is sampled. The updated and simplified version of Figure 1 is linked below:
>
> https://anonymous.4open.science/r/FairPFN-supplementary/figures/flowchart.pdf
>
> To compensate for this simplification, we've developed a pseudocode algorithm explaining our synthetic data generation process, which we propose including in the main text:
>
> https://anonymous.4open.science/r/FairPFN-supplementary/figures/data_generation.pdf

---

### Decision · Program_Chairs · 2025-05-01

**Decision:**

Accept (poster)

**Comment:**

The paper studies causal fairness in tabular data and proposes to pre-train a foundation model  (transformer) on synthetic causal fairness data, where protected attributes are modelled as binary exogenous causes. This allows mitigation of the causal effects of protected attributes using observational data without requiring prior knowledge of the underlying causal model.

The reviewers and AC find the task appealing and important to address, and the approach interesting and sound. In the initial reviews, there were some technical questions that the rebuttal was able to successfully address, as well as some minor issues (clarifications, need to discuss limitations, analysis as the complexity of the data-generating process increases etc). These should be incorporated in the final version.

After careful deliberation the decision was made to accept the paper. Congratulations to the authors! It is crucial, however, that all the clarifications and improvements promised in the rebuttal are implemented in the final version of the paper.